# The evolution of cloud microphysics upon aerosol interaction at the summit of Mt. Tai, China

Jiarong Li[1], Chao Zhu[1], Hui Chen[1,*], Defeng Zhao[1], Likun Xue[2], Xinfeng Wang[2], Hongyong Li[2], Pengfei Liu[3,4,5], Junfeng Liu[3,4,5], Chenglong Zhang[3,4,5], Yujing Mu[3,4,5], Wenjin Zhang[6], Luming Zhang[7], Kai Li[7], Min Liu[7], Hartmut Herrmann[1,2,8], Jianmin Chen[1,4,9,*]

[1]Shanghai Key Laboratory of Atmospheric Particle Pollution and Prevention (LAP3), Department of Environmental Science and Engineering, Institute of Atmospheric Sciences, Fudan University, Shanghai 200438, China

[2]Environment Research Institute, School of Environmental Science and Engineering, Shandong University, Ji'nan 250100, China

[3]Research Center for Eco-Environmental Science, Chinese Academy of Sciences, Beijing 10085, China

[4]Center for Excellence in Urban Atmospheric Environment, Institute of Urban Environment, Chinese Academy of Science, Xiamen 361021, China

[5]University of Chinese Academy of Sciences, Beijing 100049, China

[6]State Environmental Protection Key Laboratory of Urban Ambient Air Particulate Matter Pollution Prevention and Control, College of Environmental Science and Engineering, Nankai University, Tianjin 300071, China

[7]*Tai'an Municipal Ecological Environment Bureau, Shandong Tai'an Ecological Environment Monitoring Center, Tai'an 271000, China*

[8]Leibniz Institute for Tropospheric Research, Leipzig, Germany

[9]Shanghai Institute of Eco-Chongming (SIEC), No.3663 Northern Zhongshan Road, Shanghai 200062, China

*Corresponding to*: Jianmin Chen (jmchen@fudan.edu.cn) and Hui Chen (hui_chen@fudan.edu.cn)

**Abstract.** The influence of aerosols, both natural and anthropogenic, remains a major area of uncertainty when predicting the properties and behaviour of clouds and their influence on climate. In an attempt to understand better the microphysical properties of cloud droplets, the aerosol-cloud interactions, and the possible climate effect during cloud life cycles in the North China Plain, an intensive observation took place from 17 June to 30 July 2018 at the summit of Mt. Tai. Cloud microphysical parameters were monitored simultaneously with number concentrations of cloud condensation nuclei ($N_{CCN}$) at different supersaturations, $PM_{2.5}$ mass concentrations, particle size distributions and meteorological parameters. Number concentrations of cloud droplets ($N_C$), liquid water content (LWC) and effective radius of cloud droplets ($r_{eff}$) show large variations among 40 cloud events observed during the campaign. The low values of $r_{eff}$ and LWC observed at Mt. Tai were comparable with urban fogs. Clouds in clean days are more susceptible to the change in concentrations of particle number ($N_P$), while clouds formed in polluted days might be more sensitive to meteorological parameters such as updraft velocity and cloud base height. Through studying the size distributions of aerosol particles and cloud droplets, particles larger than 150 nm played important roles on forming cloud droplets with the size of 5−10 μm. In general, LWC shows positive correlation with $r_{eff}$. As $N_C$ increases, $r_{eff}$ changes from a trimodal distribution to a unimodal distribution and shift to smaller size mode. By assuming a constant cloud thickness, increase in $N_C$ and decrease in $r_{eff}$ would increase cloud albedo, which may induce a cooling effect on the local climate system. Our results contribute valuable information about regional cloud microphysics and will help to reduce the

uncertainties in climate models when predicting climate responses to cloud-aerosol interactions in North China Plain.

## 1. Introduction

Clouds are key in the atmospheric hydrological cycle, which play an important role in the atmospheric energy budget and significantly influence the global and regional climate (Chang et al., 2019;Zhang et al., 2004b). Clouds can be physically described by their liquid water contents (LWC), number concentrations of droplets ($N_C$) and effective radius of droplets ($r_{eff}$). These parameters may show small inter-annual variations for the same monitoring station (Möller et al., 1996), but they vary over a large range for different cloud types (Quante, 2004), cloud altitudes (Padmakumari et al., 2017;Zhao et al., 2018) and in different parts of a cloud (Deng et al., 2009).

The interactions between the clouds and the aerosols are complex. Clouds efficiently remove aerosols by activating CCN to cloud droplets (Croft et al., 2010;Zhang et al., 2004a). The cloud processes can incorporate large amount of fine particulate mass (Heintzenberg et al., 1989), change the size distributions (Drewnick et al., 2007;Schroder et al., 2015) and alter the CCN compositions through homogeneous and heterogeneous reactions (Roth et al., 2016). In addition, the variation of aerosol number concentrations and size distributions could alter the cloud microphysics. Through studying microphysical characteristics of cloud droplet residuals at Mt. Åreskutan, Noone et al. (1990) found that larger cloud droplets preferred to form on larger Cloud Condensation Nuclei (CCN). The aerosol-cloud interaction has been investigated for cloud processes formed under both clean and polluted conditions. Padmakumari et al. (2017) found that convective clouds over land were characterized by lower LWC and higher $N_C$ due to the perturbation of pollution aerosol. Ground-based observations by radiometers during the summers of the U.S. Mid-Atlantic region revealed that cloud events with smaller droplets (< 7 μm) occurred more frequently in the polluted years than in the clean years (Li et al., 2017b). The influence of aerosols on the cloud microphysics is evident but varies for different regions and for different cloud types.

For a given liquid water content, aerosol particles can act as CCN, lead to higher number concentrations of cloud droplets with smaller sizes and result in higher albedo (Twomey effect or first indirect effect, FIE) (Twomey, 1974). To better evaluate the influence of aerosols on the cloud microphysics, the calculation of FIE has been widely studied (Lohmann and Feichter, 2005;McComiskey et al., 2009;Twohy et al., 2005). However, the arithmetics of FIE use different parameters to represent the aerosol loading, such as the number concentration of particles, the CCN concentration and the aerosol optical depth (AOD), which makes it difficult to compare the FIE from different studies. Positive relationships between aerosol loading and $r_{eff}$, called the "anti-Twomey effect", are widely observed, especially over land (Bulgin et al., 2008;Grandey and Stier, 2010;Tang et al., 2014;Wang et al., 2014).

The increase in the aerosol concentrations results in a longer cloud lifetime, thus producing large cloud fractions (Koren et al., 2005;Albrecht, 1989), and increasing cloud top height and cloud thickness (Fan et al., 2013), which further influence

the regional and global climate (Rosenfeld, 2006;Seinfeld et al., 2016). The reduction in the precipitation or drizzle caused by the perturbation of aerosols (Andreae et al., 2004;Heikenfeld et al., 2019) delays the hydrological cycle (Rosenfeld, 2006). Through Model experiments with the Coupled Model Intercomparison Project phase 5 (CMIP5), Frey et al. (2017) found that the monthly mean cloud albedo of subtropical marine stratocumulus clouds increased with the addition of anthropogenic aerosols.

In situ measurements of cloud microphysics by aircraft or on high-altitude monitoring sites have provided some additional information for insight into the cloud processes (Allan et al., 2008;Li et al., 2017a;Padmakumari et al., 2017;Van Pinxteren et al., 2016;Reid et al., 1999). However, lacking knowledge of the size distributions of cloud droplets and aerosol particles makes it difficult to evaluate the cloud microphysics in small-scale regions (Fan et al., 2016;Khain et al., 2015;Sant et al., 2013). Discrepancy still exists between the widths of observed and simulated size distributions of cloud droplets (Grabowski and Wang, 2013). What's more, incompletely knowledge of the impact of cloud-aerosol interactions (Rosenfeld et al., 2014b), unresolved process of cloud formation (Stevens and Bony, 2013) and the lack of researches about the variation of cloud microphysical parameters at different cloud stages still hinder modelling studies.

The summit of Mt. Tai is the highest point in the centre of the North China Plain (NCP). Sufficient moisture in summer and dramatic temperature differences between day and night make it ideal for in situ orographic cloud monitoring (Li et al., 2017a). The summit of Mt. Tai is far away from anthropogenic emission sources on the ground. But high concentrations of inorganic ions in $PM_{2.5}$ (Zhou et al., 2009), abundant bacterial communities (Zhu et al., 2018), $NH_3$ and $NO_x$ emissions form biomass burning (Chang et al., 2018) have been observed at the summit, thus a strong anthropogenic influence is existing. Previous studies of cloud samples collected at the same position showed high inorganic ion concentrations (Li et al., 2017a;Wang et al., 2011), which can be attributable to the perturbation of anthropogenic aerosol. Within the present study, in situ observations at the summit of Mt. Tai were conducted and used to study the evolution of cloud microphysics upon aerosol interaction within non-precipitating clouds. Two typical cloud processes are discussed in detail to elucidate the relationship of $N_C$, $r_{eff}$ and LWC under clean or polluted conditions (indicated by $N_P$ and $N_{CCN}$) and during the cloud life cycle. The present paper provides comprehensive information for the aerosol impact on the microphysical properties of orographic clouds. The albedo based on the observed data has been estimated for climate implication.

## 2. Experiments

### 2.1. Observation duration and site

From 17 June to 30 July 2018, 40 cloud events in total were monitored at the Shandong Taishan Meteorological Station at summit of Mt. Tai (Tai'an, China; 117°13' E, 36°18' N; 1545 m a.s.l.; Fig. S1). Mt. Tai is the highest point in the central of North China Plain (NCP) and located within the transportation channel between the NCP and the Yangtze River Delta (Shen

et al., 2019). The altitude of Mt. Tai is close to 1.6 km. This height is close to the top of the planetary boundary layer in Central East China and usually sited for the characteristic of particles inputting to clouds (Hudson, 2007). Local cloud events frequently occurred at the summit of Mt. Tai, especially in summer. As shown in Fig. S2, the prevailing wind direction during this summer campaign is east wind (23.3%), southwest wind (22.8%) and south wind (21.9%), respectively. About 85.6% of wind speed was less than 8 m s$^{-1}$. While the monitored cloud events in the present study was mainly influence by south wind (34.7%) and southwest wind (22%). The arrangement of instruments was presented in Fig. S1(b).

## 2.2. Cloud microphysical parameters

A Fog Monitor (Model FM-120, Droplet Measurement Technologies Inc., USA), a forward-scattering optical spectrometer with sampling flow of 1 m$^3$ min$^{-1}$, was applied in situ for real-time displaying size distributions of cloud droplets and computing $N_C$, LWC, median volume diameter (MVD) and effective diameter (ED) in the size range of 2 to 50 μm (Spiegel et al., 2012). The corresponding equations are:

$$N_C = \Sigma N_i,$$

$$LWC = \frac{4\pi}{3} \Sigma N_i r_i^3 \rho_w,$$

$$MVD = 2 \times (\frac{\Sigma N_i r_i^3}{\Sigma N_i})^{\frac{1}{3}}$$

$$ED = 2 \times r_{eff} = 2 \times \Sigma n_i r_i^3 / \Sigma n_i r_i^2,$$

Where $N_i$ is the cloud number concentration at the ith bin, $r_i$ represents the radius at the ith bin and $\rho_w$ =1 g cm$^{-3}$ stands for the density of liquid water. Droplets are categorized into manufacture's predefined 30 size bins with sampling resolution of 1 s. The size bin widths using this configuration were 1 μm for droplets < 15 μm and 2 μm for droplets > 15 μm. The true air speed calibration and size distribution calibration of FM-120 were carried out by the manufacturer using borosilicate glass microspheres of various sizes (5.0, 8.0, 15.0. 30.0, 40.0 and 50.0 μm, Duke Scientific Corporation, USA). The difference in optical properties between the glass beads and water was taken into account during the calibration process. In this study, the sampling inlet nozzle faced the main wind direction and was horizontally set. Cloud events are defined by the universally accepted threshold values in $N_C$ and LWC, i.e., $N_C$ > 10 # cm$^{-3}$ and LWC > 0.001 g m$^{-3}$ (Demoz et al., 1996). Too short cloud events with a duration < 15 minutes were excluded.

The topography of the monitoring position could provide the vertical wind field (updraft velocity, $v_{up}$) and further affect cloud microphysical properties (Verheggen et al., 2007). Based on assumptions that air flow lines were parallel to the terrain and without occurrence of sideways convergence and divergence, $v_{up}$ was estimated by the topography of Mt. Tai and the horizontal wind speed ($v_h$) measured at the observation station (Hammer et al., 2014), the calculation equation of was:

$$v_{up} = \tan(\alpha) \times v_h$$

Where α represented the inclination angle which was estimated from the altitudes of Tai'an City and the summit of Mt. Tai

and the horizontal distance between them (Fig. S3). It should be noticed that the calculated $v_{up}$ could be considered as the upper limit of the true updraft velocity if the flow lines would not strictly follow the terrain (Hammer et al., 2014). As shown in Table S2, the averaged $v_{up}$ during two focused cloud processes (CP-1 and CP-2) studied in the present study was 0.82 m s$^{-1}$ and 0.92 m s$^{-1}$, respectively, and did not change a lot. Thus, we simply assumed that the influence of $v_{up}$ on cloud microphysical properties for CP-1 and CP-2 was the same.

In order to estimate the sampling losses due to wind speed and wind direction, the sampling efficiency (contributed by aspiration efficiency and transmission efficiency) was estimated based on the study of Spiegel et al. (2012). The sampling efficiency was depended on two parameters. One is sampling angle ($\theta_s$) which is equal to $\alpha$. The other is $R_V$ which is equal to the velocity ratio of surrounding wind speed ($U_0$) with sampling speed ($U$) of FM-120:

$$R_V = \frac{U_0}{U} = \frac{\frac{v_h}{\cos(\alpha)}}{U}$$

In the study of Spiegel et al. (2012), they calculated that the sampling efficiency under standard atmospheric conditions (p = 1013 mbar, T = 0 °C) and represented the results in their Fig. 7. Through calculation, the averaged $R_V$ of CP-1 and CP-2 was 1.02 and 1.14, respectively. Thus, we could use Fig. 7a) from Spiegel et al. (2012), where $R_V$ = 1.2, to estimate the sampling efficiency of FM-120 during CP-1 and CP-2. As can be seen, for $\theta s$ = $\alpha$ = 11.9° and 10.6° (Fig.S3), the aspiration efficiency and transmission efficiency are all close to 1. Thus, we assumed that the influences of topography and updraft velocity on Fog Monitor were small and could be ignored during CP-1 and CP-2.

### 2.3. Aerosol size distribution

A Scanning Mobility Particle Sizer (SMPS, Model 3938, TSI Inc., USA) consisting of a Differential Mobility Analyzer (DMA, Model 3082, TSI Inc., USA) and a Condensation Particle Counter (CPC, Model 3775, TSI Inc., USA) was applied to monitor the size distributions of dehumidified aerosols through a PM$_{10}$ inlet. The neutralized aerosols were classified by DMA to generate a monodisperse stream of known size according to their electrical mobility. The CPC placed downstream counts the particles and gives the number of particles with different sizes. In the present study, each scan was fixed at 5 min for every loop with a flow rate of 1.5 L min$^{-1}$ sizing particles in the range of 13.6 - 763.5 nm in 110 size bins. The mass concentrations of particles measured by SMPS (PM$_{0.8}$) was calculated from the aerosol number size distribution by simply assuming a density of $\rho$ = 1.58 g cm$^{-3}$ (Cross et al., 2007) and compared with the monitored mass concentration of PM$_{2.5}$ (Fig. 2, c). Generally, the variation of PM$_{0.8}$ and PM$_{2.5}$ were highly consistent with each other, especially when PM$_{2.5}$ was less than 20 μg m$^{-3}$. In the present study, PM$_{2.5}$ and N$_P$ (the total number concentration of aerosol particles measured by SMPS) were combined together to separate aerosol conditions of cloud processes.

### 2.4. CCN number concentration

The N$_{CCN}$ at certain supersaturations (SS) were quantified by a Cloud Condensation Nuclei Counter (Model CCN-100, DMT

Inc., USA). The CCN counter was set at five SS values sequentially for 10 min each at 0.2 %, 0.4 %, 0.6 %, 0.8 % and 1.0 % with a full scan time resolution of 50 min. Data collected during the first 5 min of each SS was excluded since the CCN counter needs time for temperature stabilization after the change of SS. The ratio of sample flow to sheath flow was set at 1:10 with a total airflow of 500 ccm. The SS of CCN counter were calibrated before the campaign and checked at the end of the campaign with monodisperse ammonium sulfate particles of different sizes (Rose et al., 2008).

## 2.5. PM$_{2.5}$ concentrations and meteorological parameters

The PM$_{2.5}$ mass concentration was measured using a beta attenuation and optical analyzer (SHARP monitor, model 5030i, Thermo Scientific Inc., USA). Meteorological parameters including the ambient temperature (T$_a$, ℃), relative humidity (RH), wind speed (WS, m s$^{-1}$) and wind direction (WD, °) were provided by Shandong Taishan Meteorological Station at the same observation point. The ground-level temperature (T$_g$), ground-level pressure (P$_g$), and dew point temperature (T$_{gd}$) were supported by National Meteorological Observatory – Tai'an Station (station number: 54827, 117°9' E, 36°9' N, 128.6 m a.s.l.).

## 2.6. Calculation of cloud base height

In the present study, the estimated lifting condensation level (LCL) is applied to represent the cloud base height (CBH) due to the lack of corresponding instruments. The calculation of LCL depends on the meteorological parameters measured at Tai'an Station. The ground-level data of temperature, dew point temperature, and pressure were used as input parameters (Georgakakos and Bras, 1984):

$$p_{LCL} = \frac{1}{(\frac{T_g - T_{gd}}{223.15} + 1)^{3.5}} \times p_g$$

$$T_{LCL} = \frac{1}{(\frac{T_g - T_{gd}}{223.15} + 1)} \times T_g$$

$$CBH = 18400 \times (1 + \frac{T_{LCL} - T_g}{273}) \times \lg\frac{p_g}{p_{LCL}}$$

Where p$_{LCL}$ is the LCL pressure; T$_{LCL}$ is the LCL temperature.

During the observation period, CBH ranged from 460.3 m to 3639.1 m with the average value of 1382.5 m. As shown in Fig. 2b, the observation station would be totally enveloped in clouds and around when cloud events occurred. The corresponding distance between the observation point and CBH was represented in Fig. 2b.

## 2.7. Calculation of FIE

The aerosol first indirect effect can be evaluated based on different cloud microphysical properties (McComiskey et al., 2009;Feingold et al., 2001). In the present study, FIE based either on the r$_{eff}$ or on N$_C$ were used calculated as

$$\text{FIE}_r = -\left(\frac{\Delta \ln r_{eff}}{\Delta \ln N_P}\right)_{LWC}, 0 < \text{FIE}_r < 0.33$$

$$FIE_N = -(\frac{\Delta \ln N_C}{\Delta \ln N_P}), \quad 0 < FIE_N < 1$$

Where $N_P$ is applied as an proxy of aerosol amount (Zhao et al., 2012; Zhao et al., 2018).

## 2.8. Calculation of cloud albedo

Cloud albedos can be calculated using the equations shown below (Seinfeld and Pandis, 2006). Assuming the cloud droplet size distribution can be approximated as monodisperse and the cloud is vertically uniform with respect to droplet size distribution (Stephens, 1978), the cloud optical thickness ($\tau_c$) could be obtained by

$$\tau_c = h(\frac{9\pi LWC^2 N_c}{2\rho_w^2})^{\frac{1}{3}}$$

Where h is the thickness of the cloud and $\rho_w$ is the density of cloud water.

For the nonabsorbing and horizontally homogeneous cloud, the two-stream approximation for the cloud albedo ($R_c$) gives as (Lacis and Hansen, 1974)

$$Albedo = \frac{\sqrt{3}(1-g)\tau_c}{2 + \sqrt{3}(1-g)\tau_c}$$

Where g is the asymmetry factor. The radius of cloud droplets was much greater than the wavelength of visible light, hence g is 0.85. The equation before becomes to

$$Albedo = \frac{\tau_c}{\tau_c + 7.7}$$

## 3. Results and discussion

### 3.1. Overview of the cloud microphysics

During 17th June to 30th July 2018, 40 cloud events were captured at the summit of Mt. Tai. The averaged $N_C$, LWC, and $r_{eff}$ of the 40 cloud events at the summit of Mt. Tai varied over the ranges of 59–1519 # cm$^{-3}$, 0.01–0.59 g m$^{-3}$ and 2.6–7.4 μm, respectively (Table S1). The monitored number concentration of cloud droplets at Mt. Tai both in the present study and in 2014 can reach 2000-3000 # cm$^{-3}$ (Li et al., 2017a), which is much higher than those values (with a range of 10–700 # cm$^{-3}$) for city fogs and convective and orographic clouds (Allan et al., 2008; Li et al., 2011; Padmakumari et al., 2017) (Table 1).

The microphysics of different clouds and fogs can generally be distinguished in a plot of $r_{eff}$ (or MVD) against LWC. As illustrated in Fig. 1, the LWC increases as the altitude increases in order of city fogs, orographic clouds and convective clouds. It is consistent with the study by Penner et al. (2004) that LWC within clouds increases linearly with altitude. The increase of LWC should be determined by the increase of $r_{eff}$ and/or $N_C$. But sometimes only one factor plays the determining role. Even though the maximum $N_C$ in Shanghai fog were higher than those in Hyderabad clouds; the larger sizes of clouds in Hyderabad determined their higher LWC values (Li et al., 2011; Padmakumari et al., 2017). When compared with previous orographic clouds, LWC at Mt. Tai appeared to show a larger range. We monitored the high values, which are comparable with convective

clouds, and the low values, which are similar to city fogs.

Different from convective clouds studied by research aircraft, orographic clouds were mainly formed in the boundary layer as air approaching the ridge, forced to rise up and cooled by adiabatic expansion (Choularton et al., 1997). Cloud events at Mt. Tai were monitored in a fixed location and more easily affected by locally transferred air mass. Therefore, it is very worthwhile to use Mt. Tai to study how the aerosol load was corresponding to a CCN influence on cloud microphysics and even the cloud life cycle.

### 3.2. Analysis on typical cloud processes

Two typical cloud processes were selected and analysed with their special characteristics. In cloud process-1 (CP-1, including one cloud event – CE-19), cloud droplets formed under a relatively stable (wind speed < 4 m s$^{-1}$) and clean (PM$_{2.5}$ ≈ 10.9 μg m$^{-3}$, N$_P$ ≈ 1425 # cm$^{-3}$) conditions accompanied by a slow increase of T$_a$ (Fig. 2, Fig. 3). During daytime, especially in the afternoon, the PM$_{2.5}$ mass concentration dramatically increased with little change in wind speed and wind direction and N$_P$ could reach to about 5000 # cm$^{-3}$ (Fig. 3). However, the perturbation of particles did not break off the cloud, which made CP-1 be the longest cloud process and persist 74 hours in the present study. Quite different from CP-1, cloud process-2 (CP-2) contained eight cloud events (CE-20 to CE-26, Fig. 3) and occurred periodically under high PM$_{2.5}$ (Fig. 2, 50.7 μg m$^{-3}$ in average) as well as high N$_P$ (Fig. 3, 1694 # cm$^{-3}$ in average) conditions. Cloud events in CP-2 formed after sunset with sharp decreasing of PM$_{2.5}$ and N$_P$, and transitorily dissipated at noon accompanied with the increase of PM$_{2.5}$, N$_P$, T$_a$ and cloud base height (CBH). For cloud water samples collected during CP-1 and CP-2, the percentage of chemical compositions did not change a lot (Fig. S4). Three dominant main anions (sulfate, nitrate and ammonia) accounted for 93.39% in CP-1 and 90.37% in CP-2 of the total measured ions. The high concentration of secondary ions in the cloud water samples indicated that clouds at Mt. Tai were dramatically influenced by anthropogenic emissions.

CP-1 was separated into four stages, including SC1 (stage-clean 1), SP1 (stage-perturbation 1), SC2 (stage-clean 2), and SP2 (stage-perturbation 2) based on whether the perturbation of particles occurred (Fig. 3b). The characteristics of SC1 and SC2 were low N$_C$ (383 # cm$^{-3}$ and 347 # cm$^{-3}$, respectively), large r$_{eff}$ (7.26 μm and 6.36 μm, respectively) and high LWC/N$_C$ (1.01 ng #$^{-1}$ and 0.75 ng #$^{-1}$, respectively, which represents averaged water each cloud droplet contained) (Fig. 3b). During SP1 and SP2, the perturbation through particles occurred. Dramatic increase of N$_C$ (949 # cm$^{-3}$ and 847 # cm$^{-3}$, respectively) and decrease of r$_{eff}$ (4.90 μm and 4.88 μm, respectively) and LWC/N$_C$ (0.35 ng #$^{-1}$ and 0.36 ng #$^{-1}$, respectively) was caused.

Each cloud event of CP-2 was separated into activation stage (S1), collision-coalescence stage (S2), stable stage (S3), and dissipation stage (S4) according to the regular changes of N$_C$ and LWC/N$_C$ (Fig. 3a). In S1, N$_C$ dramatically increased to its maximum value among the cloud events. In S2, N$_C$ declined sharply to a stable value, meanwhile LWC/N$_C$ reached the maximum value. In S3, N$_C$ was stable or slightly varied and LWC/N$_C$ started to decrease. In S4, both N$_C$ and LWC/ N$_C$ decreased sharply again and finally arrived zero. Even though the two stages (S2 and S3) in CE-25 were not totally follow the

division rules, the other six cloud events followed well. It indicated that the division was helpful to study the variations of cloud microphysical properties during CP-2. The newly formed cloud droplets during S1 were characterized by small size, high $N_C$ and low LWC/$N_C$ values (Fig. 2f and 3b). For example, about 2310 # cm$^{-3}$ of cloud droplets can quickly form in the first 2 hours of CE-20. The $r_{eff}$ of these droplets was smaller than 4.1 μm and LWC/$N_C$ was about 0.2 ng #$^{-1}$. In going from S2 to S3, the strong collision-coalescence between cloud droplets caused the increase of both $r_{eff}$ and LWC/$N_C$. In S4, the increase of PM$_{2.5}$, $N_P$ and $T_a$ (Fig. 2b and Fig. 2c) decreased cloud droplet sizes (Rosenfeld et al., 2014a), decreased the ambient supersaturation, enhanced the evaporation of small droplets (Ackerman et al., 2004), and finally caused the vanishment of cloud events (Mazoyer et al., 2019).

### 3.2.1.    Relationships among $N_P$, $N_{CCN}$ and $N_C$

During cloud processes, the relationship between Np and $N_C$ was the result of competition between aerosol particles and cloud droplets of ambient water. It depended on whether the cloud water content was sufficient, which could be reflected by the value of LWC/ $N_C$. During the studies of cloud physics, the viewpoint that the increase of $N_P$ brings more CCN and further increases $N_C$ is supported by in situ observations (Lu et al., 2007;Mazoyer et al., 2019) and modelling studies (Heikenfeld et al., 2019;Zhang et al., 2014). In contrast, Modini et al. (2015) found negative relation between $N_C$ and the number of particles with diameters larger than 100 nm due to the reduction of supersaturation by coarse primary marine aerosol particles. Some recent studies of fog also suggested that the increase of $N_P$ would decrease the ambient supersaturation and then decrease $N_C$ (Boutle et al., 2018;Mazoyer et al., 2019). Within the present study, both positive and negative relations between $N_P$ and $N_C$ have been observed. But they appeared at different cloud processes (e.g., $N_P$ and $N_C$ showed consistent variation in CP-1) and at different stages of cloud events (e.g., an obviously inverse relation between $N_P$ and $N_C$ existed in S1 and S4 while $N_P$ and $N_C$ simultaneously decreased in S2) (Figure 3a). High LWC/$N_C$ value indicating water was sufficient for new cloud droplet formation. Once $N_P$ increased, part of the cloud water was taken away by the CCN in the particles to form new droplets, and the remaining amount of water was still sufficient to maintain the previous droplets in liquid state. Positive relationship was existed between $N_P$ and $N_C$. However, lower LWC/$N_C$ values, to some extent, limited the formation of new cloud droplets. The activated particles grew at the beginning of the cloud cycle would lower the surrounding supersaturation and to some extent limit further aerosol activation (Ekman et al., 2011). The part of water taken by the CCN in the particles was not enough to active all of them to be new droplets and the remaining amount of water was also insufficient to maintain all the previous droplets in liquid state. Then the $N_C$ would decrease and the more the Np, the sharper decrease the $N_C$. Thus, the inverse relationship would be observed.

The hygroscopicity of aerosols determines the ability of aerosols acted as CCN, which can further influence cloud number concentrations. Due to the lack of corresponding instruments, the hygroscopicity parameter $\kappa$ is not available. In the study of Mazoyer et al. (2019) and Asmi et al. (2012), both of them found that high $N_{CCN}/N_P$ was associated with high $\kappa$ at a

given SS. Thus, $N_{CCN,0.2}$ ($N_{CCN}$ measured at SS = 0.2%) to $N_P$ fractions ($N_{CCN,0.2}/N_P$, CCN activation ratio) is applied to reflect the hygroscopicity of ambient aerosols at Mt. Tai. As shown in Fig. 3b $N_{CCN,0.2}/N_P$ ranged from 0.06 to 0.69 in CP-1 yet it was range from 0.22 to 0.66 in CP-2. The plot of $N_{CCN,0.2}$ versus $N_P$ was more scatter in CP-1 than that in CP-2 (Fig. 3b and Fig. 3c). Values lower than 0.22 did not appear during CP-2. Even though the settled SS in the present study (SS = 0.2%) is different from that at puy-de-Dome (SS = 0.24%), most of the data points of CP-1 and CP-2 were distributed between the two recommended dashed lines (the visually defined boundaries in within most of the data are centered, Fig. 3c and 3d) by Asmi et al. (2012). During the observation program at Puy-de-Dome, France, Asmi et al. (2012) found that higher $N_{CCN}/N_P$ and more concentrated plot of $N_{CCN,0.2}$ versus $N_P$ were usually occurred during winter when higher fraction of aged organics was observed. It indicated that the difference of aerosol organic chemical compositions during CP-1 and CP-2 might influence the $\kappa$ of aerosols and further affect the $N_{CCN}/N_P$ ratio during this two cloud processes.

### 3.2.2. Aerosol First Indirect Effect

Through studying the $FIE_r$ and $FIE_N$ of CP-1 and CP-2, it indicated that cloud droplets formed under fewer background particle numbers are more sensitive to $N_P$. As shown in Fig. 4a, except for the out-of-bound $FIE_r$ values calculated with insufficient data points when LWC was larger than 0.7 g m$^{-3}$, $FIE_r$ of 0.181–0.269 for CP-1 were always higher than those of 0.025–0.123 for CP-2 in corresponding narrow LWC ranges. We verified this with $FIE_N$. Due to the limitation of the Fog Monitor, the number of cloud droplets smaller than 2 μm may be underestimated during the activation and dissipation stages (in S1 and S4) (Mazoyer et al., 2019). Thus, only the data for S2 and S3 were employed when calculating $FIE_N$ of CP-2 (Fig. 4c). Even though the underestimation of $N_C$ may also exist in CP-1, the $FIE_N$ of CP-1 (0.544) was still higher than that of CP-2 (0.144). In the previous studies, both observation and modelling studies also found that $FIE_r$ was higher under smaller aerosol amount conditions. Twohy et al. (2005) measured the equivalent $FIE_r$ of 0.27 in the California coast while Zhao et al. (2018) used satellite observations to attribute lower values of 0.10-0.19 for convective clouds over Hebei, one polluted region in China. Using an adiabatic cloud parcel model, Feingold (2003) found that $FIE_r$ increased from 0.199 to 0.301 when $N_P$ decreased to less than 1000 # cm$^{-3}$. By using the Community Atmospheric Model version 5 (CAM5), Zhao et al. (2012) also found high $FIE_r$ values in the tropical West Pacific at Darwin (TWP) due to the low $N_P$ in December, January, and February. What's more, the perturbation of aerosol particles would cause stronger albedo enhancements when pollution is low in the ambient air (Platnick et al., 2000). Through studying the impact of ship-produced aerosols on the microstructure and albedo of warm marine stratocumulus clouds, Durkee et al. (2000) found that the clean and shallow boundary layers would be more readily perturbed by the addition of ship particle effluents. In addition, the meteorological conditions and the topography during the monitoring period would also affect the microphysical properties of clouds. The sensitivity analysis of $N_C$ to CBH and $v_{up}$ was estimated by applying the equation as $S(X_i)=\partial \ln N_C/\partial \ln X_i$, where $X_i$ represented CBH and $v_{up}$. As shown in Talbe S2, CP-2 was more sensitive to the variation of meteorological parameters if compared with CP-1. It was consistent with the study of

McFiggans et al. (2006). They found that the sensitivity of $N_C$ to $v_{up}$ increased while the sensitivity of $N_C$ to $N_P$ decreased when $N_P > 1000 \# cm^{-3}$. In the present study, the higher values of $FIE_r$ and $FIE_N$ of CP-1 indicated that if the same amount of aerosol particles entered the cloud, the size of cloud droplets in CP-1 would decrease more than that in CP-2. The albedo during CP-1 would be more susceptible to the change of aerosol particles. While the higher values of S(CBH) and S($v_{up}$) of CP-2 indicated that CP-2 was more sensitive to the change of CBH and $v_{up}$. It might cause the periodical variations of cloud microphysical properties during CP-2.

The positive $FIE_r$ and $FIE_N$ at Mt. Tai mean that the increase in $N_P$ are accompanied by decreased $r_{eff}$ and increased $N_C$. No negative $FIE_r$ were found in the present study. Yuan et al. (2008) and Tang et al. (2014) applied AOD to represent aerosol loading and found negative $FIE_r$ (indicating $r_{eff}$ increased with the increasing of AOD) near coastlines of the Gulf of Mexico, the South China Sea and over Eastern China with the surrounding sea. By using the 2-D Goddard Cumulus Ensemble model (GCE), Yuan et al. (2008) explained that the positive relationship between $r_{eff}$ and AOD appeared to originate from the increasing slightly soluble organics (SSO) particles. The increase of SSO would act to increase of the critical supersaturation for particles to be activated and resulted in less numbers of activated particles. With Moderate Resolution Imaging Spectroradiometer (MODIS) observations, Tang et al. (2014) explained that the negative FIE values were likely attributable to meteorological conditions from the South and Southeast China, which usually favoured transport of both pollutants and water vapour and led to simultaneous increases in both AOD and $r_{eff}$. Compared with these regions, the summit of Mt. Tai is relatively far from the sea (around 230 km from the Bohai Sea and Yellow Sea) (Guo et al., 2012). The air brought aerosols but with less moist. It might hinder the growth of cloud droplets and caused the negative relation between $N_P$ and $r_{eff}$. An increase in LWC might reduce the FIE, especially at coastal sites (McComiskey et al., 2009;Zhao et al., 2012). However, weak variations of $FIE_r$ with an increase of LWC were found at Mt. Tai (Fig. 4a). It may be due to the high aerosol loading during cloud processes (Zhao et al., 2012).

### 3.2.3.  Size distribution of cloud droplets and particles

To illustrate the evolution of the aerosol particles and the cloud droplets during the cloud processes, the size distributions of $N_P$ and $N_C$ during different cloud stages are plotted in Fig. 5. For each of the four size bins ranged from 2 to 13 μm, cloud number concentrations of SC1 and SC2 were lower than those of SP1 and SP2. In the size bin of 13–50 μm, however, $N_C$ of SC1 and SC2 were the largest (Fig. 5b). This size distributions of cloud droplets in SC1 and SC2 resulted in the larger $r_{eff}$ during the two stages, which was consistent with the result shown in Fig. 3b. During two perturbation stages of SP1 and SP2 in CP-1, the numbers of aerosol particles in all size bins increased. But the increase of aerosol particles larger than 150 nm was the smallest, indicating that aerosols larger than 150 nm were more easily activated into cloud droplets. The activation of aerosol particles with the size larger than 150 nm in the present study dramatically increased $N_C$ of 5–10 μm and made $N_C$ of SP1 and SP2 in different size bins all comparable with those of CP-2 (Fig. 5b).

As shown in Fig. 5c, cloud droplets with $D_C$ ranging from 5 to 10 μm had high $N_C$ in each stage in CP-2 and cloud droplets with $D_C$ ranging from 13 to 50 μm had low $N_C$ in each stage if compared to CP-1. It caused the lower $r_{eff}$ in CP-2 than CP-1. During CP-2, aerosol particles with diameters larger than 150 nm quickly decreased by activation when cloud events occurred, while the number of aerosol particles in the size of 50-150 nm were slightly influenced by cloud events (the first panel of Fig. 5a). It was consistent with the study of Targino et al. (2007) who found aerosol size distributions of cloud residuals, which represented aerosol particles activated to cloud droplets, peaked at about 0.15 μm at Mt. Åreskutan. Mertes et al. (2005) also found that particles centered at $d_p$ = 200 nm could be efficiently activated to droplets while most Aitken mode particles remained in the interstitial phase. Compared with other stages, S1 had the highest $N_C$ in three size bins of [2, 5) μm and [5, 7) μm. It indicated that large numbers of cloud droplets with small sizes were formed in the beginning of cloud events in CP-2.

**3.3. Relations among LWC, $R_{eff}$ and $N_C$**

The 5 min averaged LWC for CP-1 and CP-2 is plotted against corresponding $r_{eff}$ in Fig. 6a. Large cloud droplets ($r_{eff} > 8$ μm) were observed in CP-1, while the $r_{eff}$ for CP-2 varied narrowly in the range of 2.5–8 μm.

Cloud droplets with $r_{eff} > 8$ μm only occurred in the two relatively clean stages, SC1 and SC2, during CP-1. It was due to the weaker competition among droplets at lower $N_{CCN}$ conditions. This has also been observed in the U.S. Mid-Atlantic region where cloud droplets with larger sizes are more easily formed with lower $N_{CCN}$ (Li et al., 2017b). At the same LWC level, the growth of cloud droplets during SP1 and SP2 was obviously limited if compared with SC1 and SC2, which is referred to as the "Twomey effect" (Twomey, 1977). This is consistent with the illustration in Fig. 3 that cloud droplets in SP1 and SP2 were smaller.

The variation of LWC was determined by the change of $r_{eff}$ and/or $N_C$. However, the decisive factor may be different in different stages of the cloud. As shown in the lower panel of Fig. 6a, CE-20 was taken as an example to discuss the relation among LWC, $R_{eff}$ and $N_C$ in different cloud stages. During S1, the existing numerous CCN (Fig. 3a) were quickly activated to form cloud droplets. The newly formed droplets are characterized with small sizes but large numbers. They will suppress the beginning of collision-coalescence processes (Rosenfeld et al., 2014a) and may further significantly delay raindrop formation Qian et al. (2009). In S1, positive relation existed between $N_C$ and $r_{eff}$. Both the increase in $N_C$ (from 1188 # cm$^{-3}$ to 2940 # cm$^{-3}$) and the growth of $r_{eff}$ (from ~3.5 μm to ~4.5 μm) boosted the LWC in this stage. This is different from Mazoyer et al. (2019)'s result that they found a clearly inverse relationship between the number and the size of droplets at the beginning of the first hour of fog events during the observation in suburban Paris. When compared with fog, cloud is usually formed under conditions with more condensible water vapour (Fig. 1). The limited growth of droplets in fog will not occur in cloud. It caused the positive relationship with cloud droplet number and droplet size. At the beginning of S2, $N_C$ reaches the maximum. The high $N_C$ yields a great coalescence rate between cloud droplets. Meanwhile, the coalescence process is self-accelerating (Freud and Rosenfeld, 2012) and thus causes the quick decrease of $N_C$ (Fig. 3a). This makes cloud droplets in S2 characterized by larger

sizes as well as lower number concentrations, whilst LWC simply varies in a relatively narrow range (Fig. 6a). During S3, $N_C$ is almost constant due to the formation, coagulation, and evaporation of the cloud droplets reaching a balance. As shown in the panel, the relationship between $r_{eff}$ and LWC in this stage could be fitting as $r_{eff}=a \times LWC^{0.34 \pm 0.02}$, which means under the increase of LWC, the $N_C$ was almost unchanged. The variation of LWC values is mainly due to the changes of droplet sizes. At the dissipation stage of S4, the clouds vanish due to mixing with the dry ambient air (Rosenfeld et al., 2014a). The previously activated CCN returned back to the interstitial aerosol phase due to the evaporation of the droplets (Verheggen et al., 2007). Both $N_C$ and $r_{eff}$ decline. It also illustrates in Fig. 5c that all the $N_C$ of the five size bins of cloud droplets decrease in S4.

In order to investigate the variation of $r_{eff}$ upon $N_C$, the distribution of $r_{eff}$ was classified with different $N_C$ ranges in Fig. 6b. For $N_C < 1000$ # $cm^{-3}$, $r_{eff}$ displayed a trimodal distribution and concentrated on 3.25 μm (Peak-1), 4.86 μm (Peak-2) and 7.52 μm (Peak-3), respectively. Peak-1 corresponded to cloud droplets with low $N_C$, LWC, and $r_{eff}$ values while the $N_{CCN0.2}$ was very high (Fig. 6c). These points represented cloud droplets in the incipient stage or the dissipation stage of cloud events where large numbers of CCN exist in the atmosphere. Peak-2 and Peak-3 represented the mature stages for cloud events with different environmental conditions. Peak-3 represented cloud droplets formed under a relatively cleaner atmosphere. In this circumstance, CCN were efficiently activated and had a lower concentration remaining in the atmosphere (Fig. 6c). The sufficient ambient water vapour accelerated the growth of the formed droplets, which were characterized with low $N_C$ and LWC but large $r_{eff}$. Peak-2 represented cloud droplets formed under relatively polluted conditions and was the only peak found for $N_C$ larger than 1000 # $cm^{-3}$. With the increase of $N_C$, the distribution of this peak narrowed and slightly moved to lower $r_{eff}$ mode.

The thickness of orographic cloud was usually very thin (Welch et al., 2008). If assuming the cloud thickness during CP-1 and CP-2 were equal, albedo would depend on the values of LWC and $N_C$ as described in Section 2.8. Cloud albedo during CP-2 was always higher than that during CP-1, especially when the cloud thickness was lower than about 2500 m (Fig. 6d). Through studying marine stratocumulus clouds in the north-eastern Pacific Ocean, Twohy et al. (2005) also found that the increase of $N_C$ by a factor of 2.8 would lead to 40% increase of albedo going from 0.325 to 0.458. It indicated that the higher $N_C$ would increase the cloud albedo if assuming no change of cloud thickness.

## 4.  Conclusion

From 17 June to 30 July 2018 in-situ observations of number concentrations and size distributions of aerosol particles and cloud droplets are employed to show aerosol-cloud interactions at the summit of Mt. Tai. Large variations of the characteristic values in terms of $N_C$, LWC and $r_{eff}$ were found during the observation period. Compared with other orographic clouds, droplets with smaller $r_{eff}$ and lower LWC exist at Mt. Tai, which are comparable with urban fogs.

Two typical cloud processes, CP-1 and CP-2, are applied to study the cloud-aerosol interactions based on the aerosol

characteristics (especially $N_P$ and $N_{CCN}$) before cloud onsets. For the CP-1, which corresponded to relatively clean conditions, water content is sufficient while $N_{CCN}$ limits cloud droplet formation. The newly formed cloud droplets are characterized with low $N_C$ but high LWC/$N_C$ and large $r_{eff}$. When particle perturbation occurs, $N_C$ dramatically increased by about three times. Large numbers of $N_{CCN}$ will compete for the system water content with the formed cloud droplets and, as a result, further dramatically decrease the LWC/$N_C$ and $r_{eff}$ values of cloud droplets. In CP-2, $N_P$ before the cloud onset is high and $N_{CCN}$ is sufficient. Water vapour becomes the limitation for cloud formation. Large numbers of small cloud droplets with low LWC/$N_C$ formed in the incipient stage of cloud events. In addition, periodically changes of cloud microphysical properties were found. Both positive and negative relations between $N_P$ and $N_C$ have been observed in the present study, which depended on the values of LWC/$N_C$.

Both positive $FIE_r$ and $FIE_N$ values at Mt. Tai indicate that the increase of $N_P$ will decrease $r_{eff}$ and increase $N_C$ of cloud droplets. $FIE_r$ and $FIE_N$ values are lower with higher $N_P$ and $N_{CCN}$. This represents that the increase of $N_P$ will more strongly decrease the size and increase the number of cloud droplets under the conditions of smaller aerosol amount. Through studying the size distributions of aerosol particles and cloud droplets, higher $N_C$ in the size bin of 13–50 μm resulted in the larger $r_{eff}$ during the two clean stages in CP-1. When perturbation of aerosol particles occurred, particles larger than 150 nm can be efficiently activated to cloud droplets and make important contributions to the increase of cloud droplets in the size range of 5–10 μm.

The LWC of cloud depended on the change of $r_{eff}$ and $N_C$. However, the decisive factor may differ at different stages of the cloud. In general, the $r_{eff}$ of cloud droplets correlates positively with LWC. But in different $N_C$ ranges, the $r_{eff}$ of cloud droplets show different distribution shapes. For $N_C < 1000 \# cm^{-3}$, $r_{eff}$ displayed a trimodal distribution. Three peaks were 3.25 μm, 4.86 μm and 7.52 μm, respectively. With the increase of $N_C$, a narrowed unimodal distribution of $r_{eff}$ appeared and the peak value slightly moved towards lower $r_{eff}$ mode. For a constant cloud thickness, the increased $N_C$ and decreased $r_{eff}$ dramatically increase the cloud albedo, which may further influence the regional climate in the North China Plain.

The local topography of the surrounding areas at Mt. Tai supplies a potential access for aerosol transportation and can affect the measured cloud droplet distributions by increasing turbulence or causing orographic flows. Even though the summit of Mt. Tai is far away from the polluted sources, the transported CCN could change the cloud microphysical properties (i.e., during CP-1). The cloud microphysical parameters derived in our study characterized the cloud features in the North China Plain, and provided valuable data for modelling studies of cloud microphysics in the future.

**Data availability**

All data used to support the conclusion are presented in this paper. Additional data are available upon request. Please contact the corresponding authors (Jianmin Chen (jmchen@fudan.edu.cn) and Hui Chen (hui_chen@fudan.edu.cn)).

**Author contribution.**

JC, HC conceived the study. JL and CZ performed the field experiments and sampled cloud water. JL analysed the data and wrote the main manuscript text. JC, HC, DZ, CZ and HH revised the initial manuscript. LX, XW and HL supported the meteorological data and $PM_{2.5}$ mass concentration. PL, JL, CZ, YM and WZ assisted in instrument maintenance. LZ,

5    KL and ML contributed to the organization and arrangement of the field observation. LZ provided the meteorological parameters of Tai'an City. All of the authors discussed the results, and contributed to the final manuscript.

**Competing interests.**

The authors declare no conflict of interest.

**Acknowledgement**

10    This work was supported by the Ministry of Science and Technology of China (2016YFC0202700), Tai'an Research Project (SDTASJ2018-0761-00), National Natural Science Foundation of China (91843301,91743202, 41805091, 21806020),and Marie Skłodowska-Curie Actions (690958-MARSU-RISE-2015).

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

**Tables and Figures**

**Table 1: Comparison of clouds monitored at Mt. Tai with city fogs, convective clouds monitored by research aircrafts and other orographic clouds. Including sampling information (site, period and altitude), the range of PM$_{2.5}$ mass concentrations, the range of microphysical parameters (number concentrations of cloud droplets-N$_c$, liquid water content-LWC, median volume diameter-MVD, effective radius-r$_{eff}$) and the number of monitored clouds/cloud events/fog events.**

| Sampling Site | Period | Altitude (m a.s.l) | PM$_{2.5}$ ($\mu$g m$^{-3}$) | N$_C$ (# cm$^{-3}$) | LWC (g m$^{-3}$) | MVD ($\mu$m) | r$_{eff}$ ($\mu$m) | Number of clouds/cloud events/fog events | Reference |
|---|---|---|---|---|---|---|---|---|---|
| **City Fog** | | | | | | | | | |
| Shanghai, China | Nov. 2009 | 7 | - | 11-565 | 0.01-0.14 | 5.0-20.0 | - | 1 | (Li et al., 2011) |
| Nanjing, China | Dec. 2006- Dec. 2007 | 22 | 0.03[a]-0.60[a] | - | 2.69e$^{-3}$-0.16 | - | 1.6[b]-2.7[b] | 7 | (Lu et al., 2010) |
| **Convective Clouds** | | | | | | | | | |
| Amazon Basin/cerrado reCompagions, Brazil | Aug.-Sept. 1995 | 90-4000 | - | - | 0[d]-2.10[d] | - | 2.8[d]-9.2[d] | >1000 | (Reid et al., 1999) |
| Hyderabad - The Bay of Bengal, India | 29[th] Oct. 2010 | 1300-6300 | | 10[d]-380 | 0[d]-1.80 | | 3.8[d]-17.0 | 1 | (Padmakumari et al., 2017) |
| **Orographic clouds** | | | | | | | | | |
| Mt. Schmücke, Germany | Sep.-Oct. 2010 | 937 | - | - | 0.14-0.37 | - | 5.7-8.7 | 8 | (Van Pinxteren et al., 2016) |
| East Peak Mountain, Puerto Rico | Dec. 2004 | 1040 | - | 193-519 | 0.24-0.31 | 14.0-20.0 | - | 2 | (Allan et al., 2008) |
| Mt. Tai, China | Jul.-Aug. 2014 | 1545 | 11.1-173.3 | 4-2186 | 0.01-1.52 | 1.6-43.0 | 0.8-18.9 | 24 | Unpublished data from (Li et al., 2017a) |
| Mt. Tai, China | Jun.-Jul. 2018 | 1545 | 1.2-127.1 | 10-3163 | 1.01e$^{-3}$-1.47 | 4.4-25.0 | 2.4-13.4 | 40 | This study |
| Mt. Tai, China (CP-1[c]) | 10[th] – 13[th] Jul. 2018 | 1545 | 1.3-40.7 | 11-2470 | 1.12e$^{-3}$-1.47 | 4.6-17.4 | 2.5-10.7 | 12 | This study |
| Mt. Tai, China (CP-2[c]) | 13[th] – 20[th] Jul. 2018 | 1545 | 1.2-66.2 | 10-3163 | 1.03e$^{-3}$-1.10 | 4.6-13.5 | 2.4-7.9 | 12 | This study |

[a] Represents the mass concentrations of PM$_{10}$. [b] Represents the range of averaged radium. [c] Two cloud processes which are detailedly discussed in this study. [d] Values were read from the graphs.

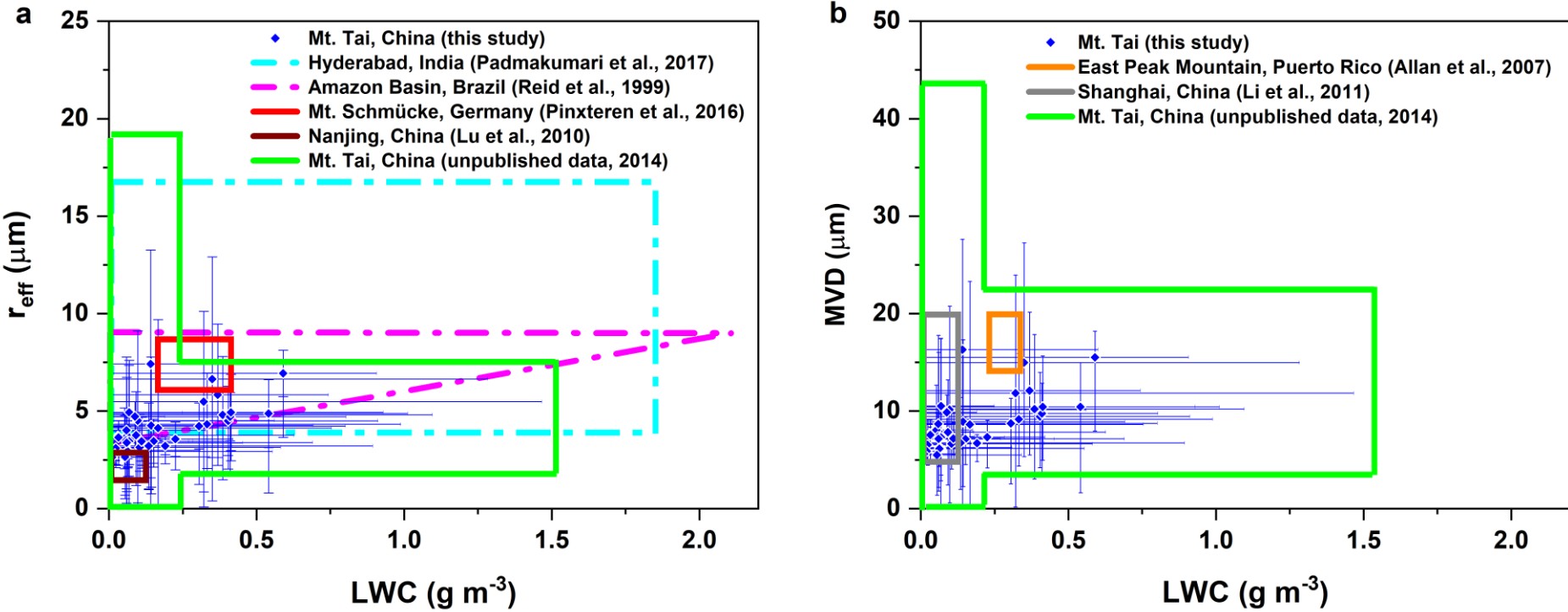

**Figure 1: Plots of effective radius (r$_{eff}$, a) or medium volume diameter (MVD, b) against liquid water content (LWC) for clouds and fogs from the literatures. The dashed and solid shapes indicated the airborne and land observation, respectively. The blue diamonds with error bars represented the average LWC and r$_{eff}$ (or MVD) of 40 cloud events observed at Mt. Tai in the present study with corresponding ranges**

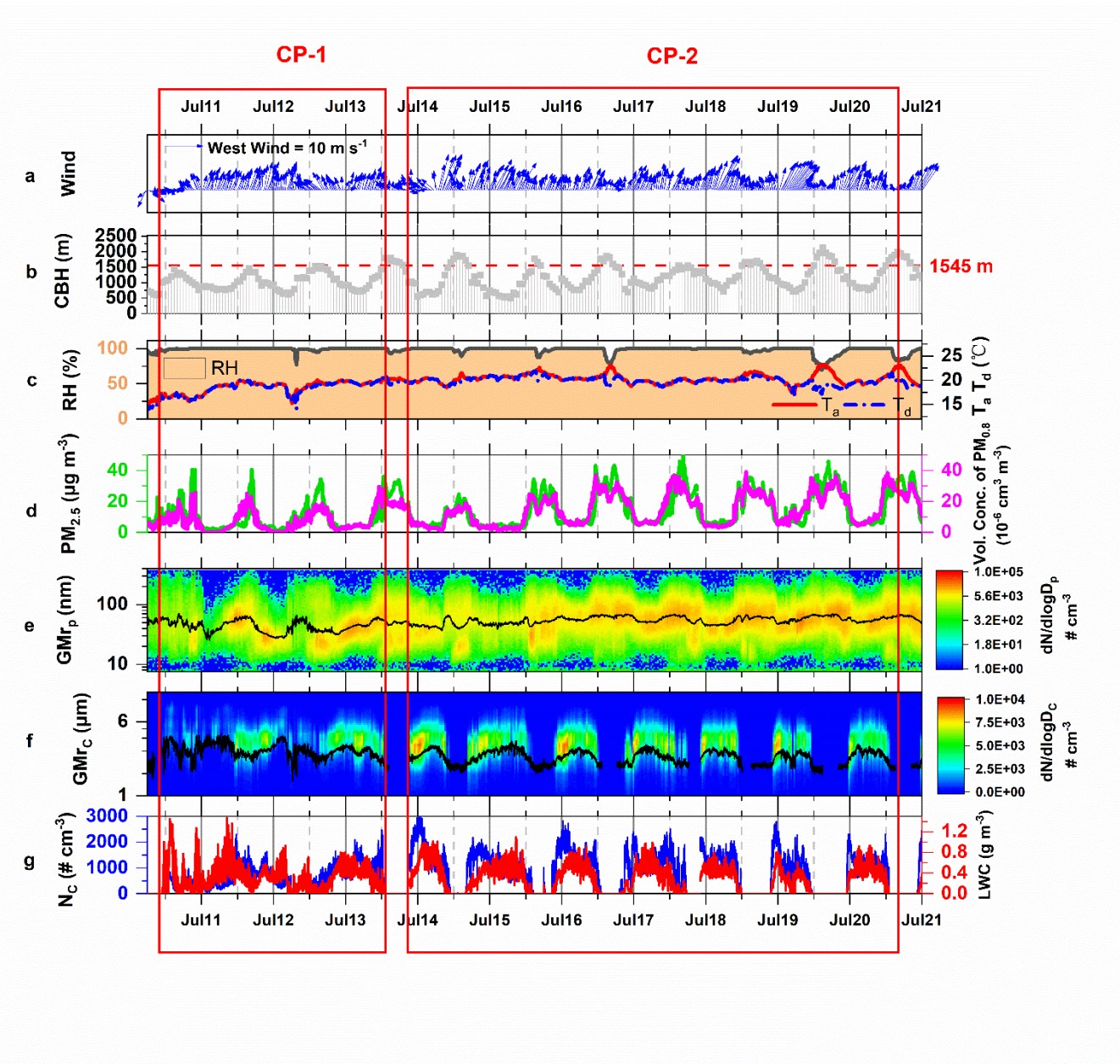

**Figure 2: The monitoring information of CP-1 and CP-2. Including (a) Wind speed (WS, m s$^{-1}$) and wind direction (WD), (b) cloud based height (CBH, m) (c)relative humidity (RH, %), ambient temperature (T$_a$, ℃) and dew point temperature (T$_d$, ℃) (d) PM$_{2.5}$ mass concentrations (μg m$^{-3}$) and volumn concentration of PM$_{0.8}$ (10$^{-6}$ cm$^3$ cm$^{-3}$) (e) size distribution of particles (13.6-763.5 nm) and corresponding geometric mean radius (GMr$_P$) (f) size distribution of cloud droplets (2-50 μm) and corresponding geometric mean radius (GMr$_C$) (g) N$_C$ and LWC of cloud droplets.**

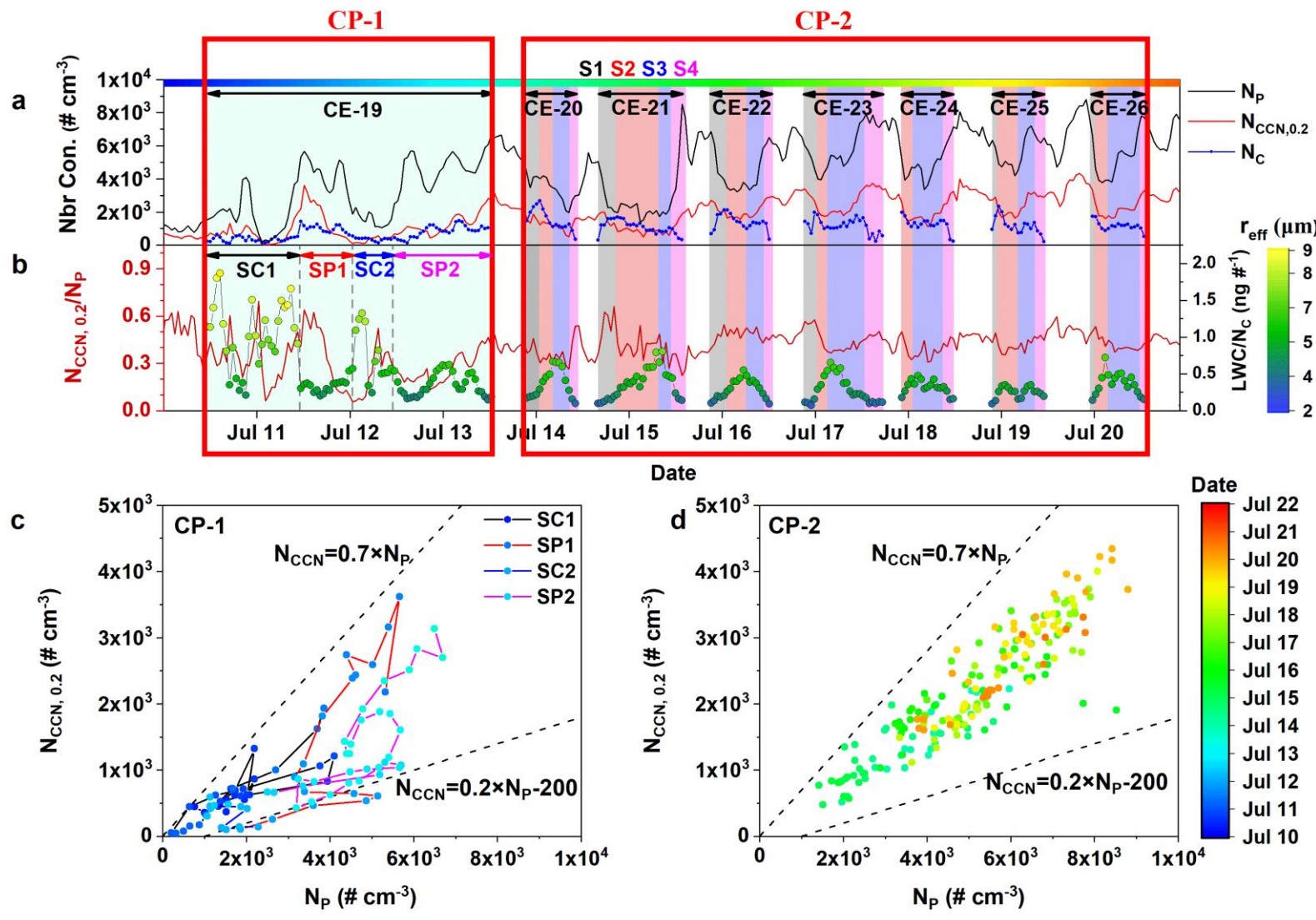

**Figure 3: Variation of (a) $N_C$, $N_P$ and $N_{CCN,0.2}$ (b) $N_{CCN,0.2}/N_P$ and LWC/$N_C$ during CP-1 and CP-2. The plot of $N_{CCN,0.2}$ versus $N_P$ (c) in CP-1 (d) in CP-2. The two dashed lines are the visually defined boundaries from the study of Asmi et al. (2012).**

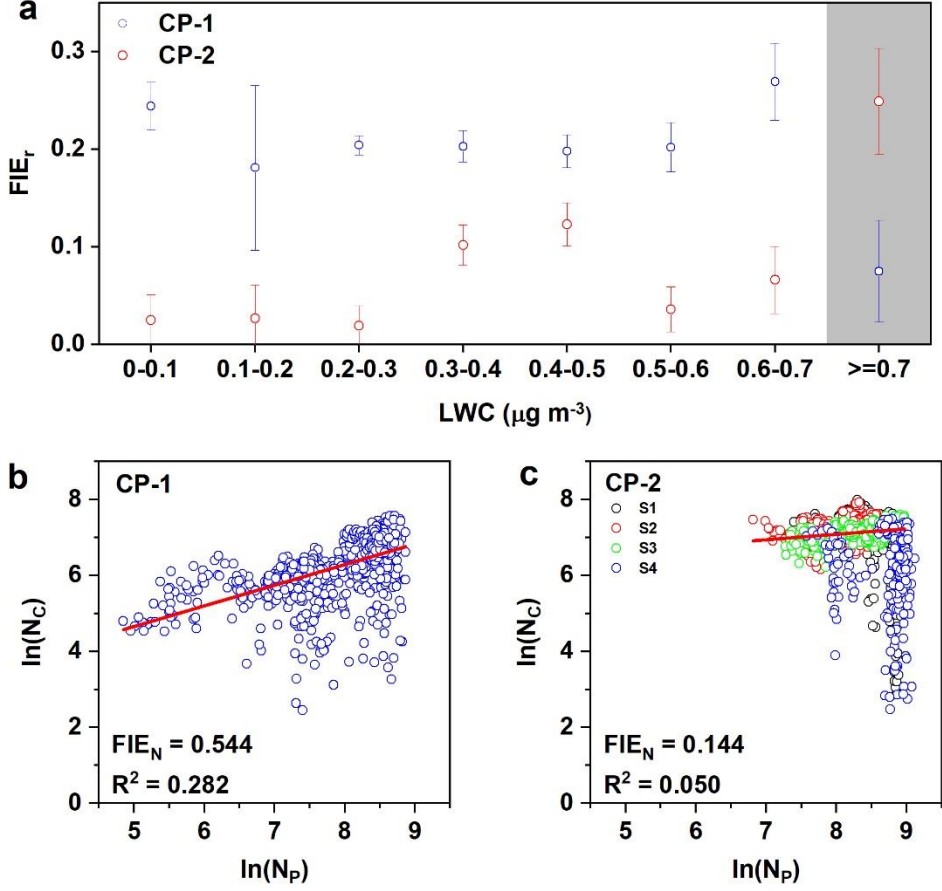

**Figure 4: The determination of FIE (a) based on r$_{eff}$ (b) and (c) based on N$_C$.**

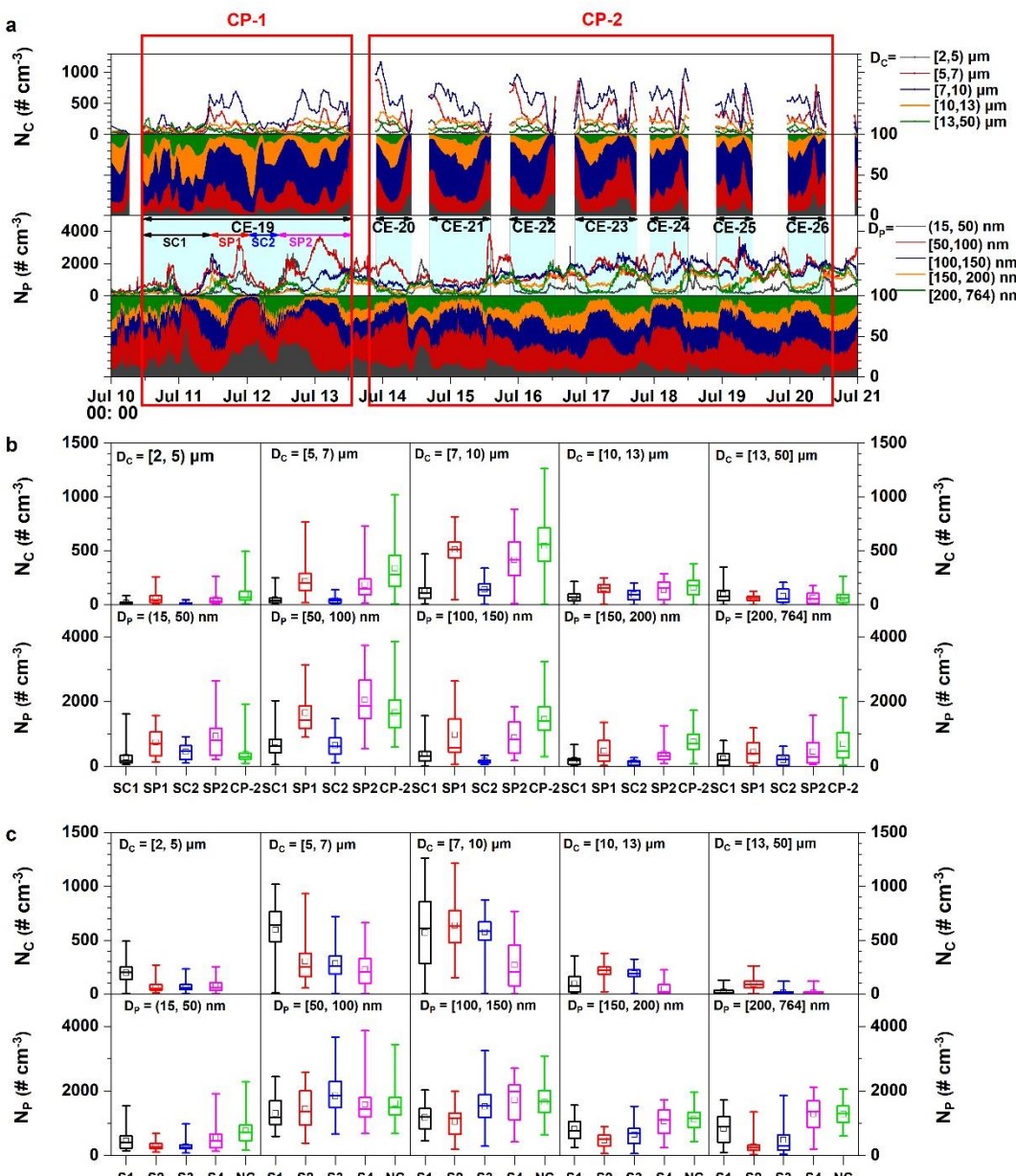

**Figure 5: Size distribution of particles and cloud droplets during CP-1 and CP-2. (a) Time series plot of $N_C$ in five size ranges ([2, 5) μm, [5, 7) μm, [7, 10) μm, [10, 13) μm and [13, 50) μm) and $N_P$ in five size ranges ((15, 50) nm, [50, 100) nm, [100, 150) nm, [150, 200) nm, [200, 765) nm). (b) five size ranges of $N_C$ and five size ranges of $N_P$ in SC1, SP1, SC2, SP2 and CP-2 (c) five size ranges of $N_C$ and five size ranges of $N_P$ in S1, S2, S3 ,S4 and NC ("NC" in (c) represents particle size distributions during cloudless period).**

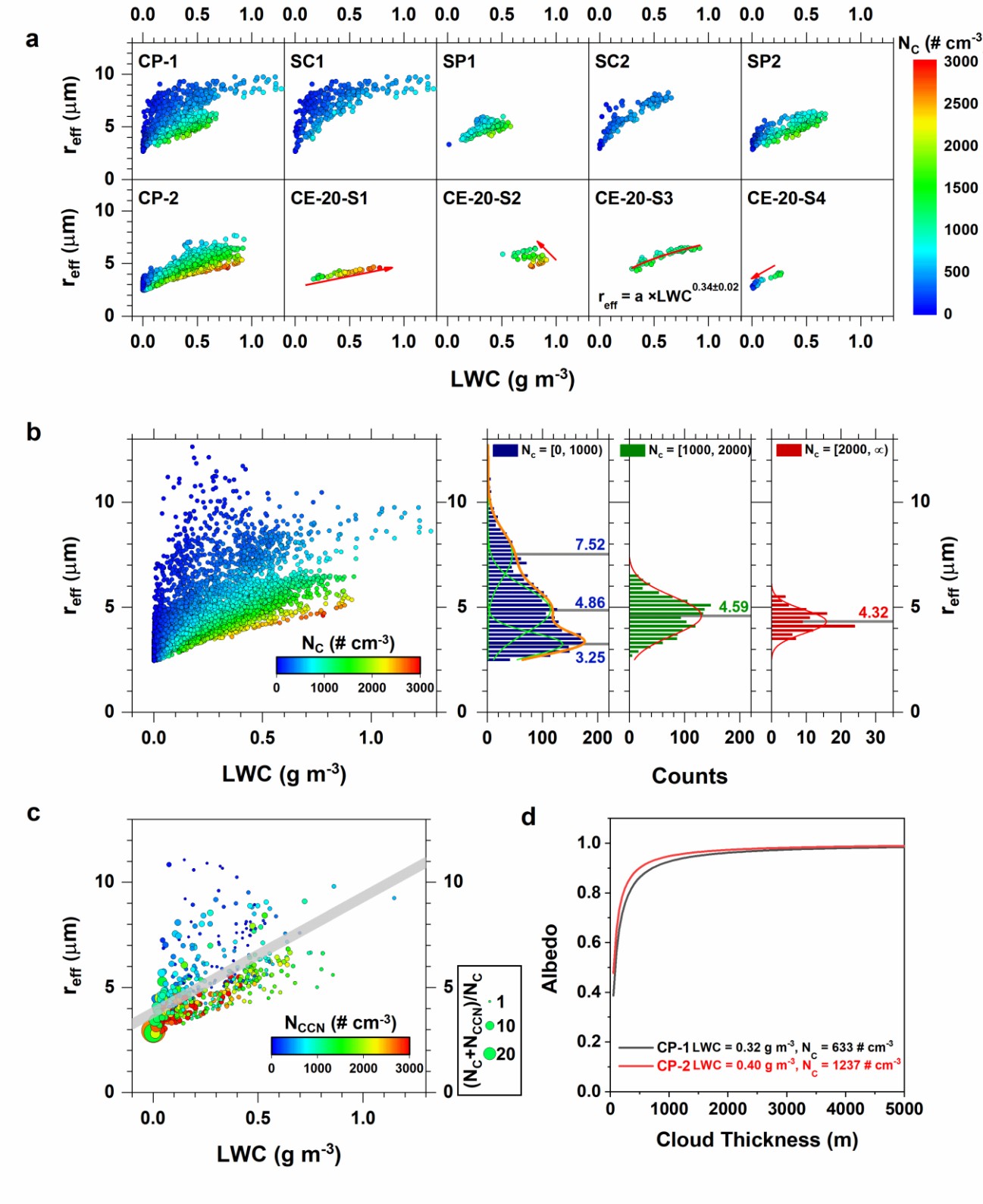

**Figure 6: The plot of LWC versus $r_{eff}$ (a) in different cloud stages of CP-1 and CP-2 (b) under different $N_C$ ranges (c) under different $N_{CCN}$. The time resolution of the corresponding data was 5 min in (a), (b) and 50 min in (c). (d) The plot of albedo versus the variation of cloud thickness during CP-1 and CP-2. The averaged values of LWC and $N_C$ of CP-1 and CP-2 were applied to calculate albedo according to the equations in Section 2.8.**