# Peer review of "The evolution of cloud and aerosol microphysics at the summit of Mt."

_Atmospheric Chemistry and Physics, 2019_

## Referee Comment (RC1) · Anonymous Referee #1 · 12 Oct 2019

General comments: This study investigates aerosol-cloud-interactions (ACI) using measurements from the high mountain site of Mt. Tai in China. As limited studies of ACI exist from high altitude measurement stations in this region, the study can potentially provide some useful data about these complex processes to the scientific community. However, the methodologies employed within this manuscript to investigate ACI are questionable, and lacking the necessary in-depth analysis currently associated with probing ACI - one of the most challenging topics currently facing the climate community. A number of conclusions presented are unsupported by the data, and rather arbitrary in nature. Numerous statements throughout the manuscript are not persuasive or lack evidence. Furthermore, the manuscript is not organised very well and the language and grammar throughout is far from the quality required for a scientific

publication. Given the large concerns associated with some of the methodologies applied, subsequent conclusions drawn, and the general quality of the text I recommend a major revision of the entire manuscript before consideration for publication.

Major comments

Methodology:

When investigating ACI it is of crucial importance to separate the contributions of changes in both aerosol and meteorology on any observed or simulated cloud response; as performed in other studies of ACI in the scientific literature, e.g. Malavelle et al., 2017. There does not appear to have been any attempt to account for variations in the meteorology in this study. If any of the analysis related to ACI is to remain in the manuscript, the study should include, but not limited to, the following additional analysis:

A. A detailed description of the measurement station with regard to location of instruments and prevailing meteorology. A picture and/or schematic is required to put the results in context of the environment in which they were measured, in particular, statistics on the height of measurements in relation to cloud base/top.

B. Isolating the role of below cloud variations in meteorology, (updraft velocity) on in-cloud variations in supersaturation and cloud microphysical properties, e.g. cloud liquid water content; cloud droplet effective radius, cloud droplet number concentration. There is a vast amount of literature addressing this, e.g. Lance et al., 2004. If observations of cloud base updraft are not available, then alternative approaches should be sought, e.g. using a cloud model in conjunction with the in-cloud measurements to probe sensitivity to variations in meteorology in a robust manner.

C. Accounting for the role of measurement height relative to cloud base in analysis. The measured cloud microphysical properties will be strongly dependant at the height they are measured in relation to cloud-base. This needs to be accounted for in the data

analysis prior to drawing conclusions regarding the role of aerosols on measured cloud properties.

D. Accounting for the role of topography on cloud droplet formation (Romakkaniemi et al., 2017).

E. A more robust isolation of anthropogenic pollution using either air-mass back trajectory based approaches such as Tunved et al., 2013, or chemical composition analysis if available. Furthermore, PM2.5 is not the appropriate measurement to separate aerosol conditions to investigate ACI. Please use an appropriate measure of the aerosol physical properties.

F. A discussion of the role of wind direction on the reliability of measurements of cloud properties from the Fog monitor, see discussion on cloud droplet measurements in Leskinen et al., 2009 as well as a detailed description of any corrections performed to the measured parameters and uncertainties associated with sampling methods.

G. Justification of choice of metrics, e.g. CCN at 0,2% supersaturation and others not commonly employed in ACI process studies, e.g.: Nccn(0.2)/Np. Why did you not focus on the droplet activated fraction (Nc/Np)?

H. In light of the new analysis associated with (A-G) a newly revised, clear explanation of cloud processes using the observations. Conclusions should not be presented as fact unless they are fully supported by the observations in the manuscript.

I. A detailed explanation of how cloud top albedo was calculated including any assumptions made and a discussion as to their validity. An assumption is made related to the calculation of cloud liquid water path, e.g. Stephens, 1978 that is not discussed. Is the assumption of 100m cloud depth valid? Furthermore, it appears that this calculation might be inconsistent as cloud top measurements of cloud microphysical properties are not provided.

Given the strong reservations on large elements of the methodology employed for the

scientific analysis in the study I recommend that these are addressed in full in the first instance prior to consideration for publication.

Regarding the terminology, language and grammar: This needs improving throughout the manuscript. Sentences found throughout such as: Line 24:25: "the perturbation of particles rapidly scrambled the water of the formed cloud droplets" Do not meet the quality required for scientific publication. In addition, throughout the manuscript some ambiguous and unclear terminology is employed. I strongly recommend that if the major revisions regarding methodology can be addressed that the manuscript text be then carefully revised, probably by a native English speaker.

References:

Lance, S., Nenes, A., and Rissman T. A.: Chemical and dynamical effects on cloud droplet number: Implications for estimates of the aerosol indirect effect, J. Geophys. Res., 109, D22208, doi:10.1029/2004JD004596, 2004.

Leskinen, A., Portin, H., Komppula, M., Miettinen, P., Arola, A., Lihavainen, H., Hatakka, J., Laaksonen, A., and Lehtinen, K. E. J.: Overview of the research activities and results at Puijo semiurban measurement station, Boreal Environ. Res., 14, 576–590, 2009.

Malavelle FF, Haywood JM, et al.:. Strong constraints on aerosol-cloud interactions from volcanic eruptions, Nature, volume 546, no. 7659, pages 485-491, 2017.

Romakkaniemi, S., Maalick, Z., Hellsten, A., Ruuskanen, A., Väisänen, O., Ahmad, I., Tonttila, J., Mikkonen, S., Komppula, M., and Kühn, T.: Aerosol–landscape–cloud interaction: signatures of topography effect on cloud droplet formation, Atmos. Chem. Phys., 17, 7955–7964, https://doi.org/10.5194/acp-17-7955-2017, 2017.

Stephens, G. L. Radiation profiles in extended water clouds. II: parameterization schemes. J. Atmos. Sci. 35, 2123–2132, 1978.

Tunved, P., Ström, J., and Krejci, R.: Arctic aerosol life cycle: linking aerosol size

distributions observed between 2000 and 2010 with air mass transport and precipitation at Zeppelin station, Ny-Ålesund, Svalbard, Atmos. Chem. Phys., 13, 3643-3660, https://doi.org/10.5194/acp-13-3643-2013, 2013.

---

## Referee Comment (RC2) · Kevin Noone (Referee) · 21 Nov 2019

General Comments

The authors present very interesting measurements made at Mt. Tai in northeast China. They have a useful set of measurements from an under-represented region. As such, it would be valuable for these data to become available. However, I feel that two factors prevent the manuscript from being published in its current form: 1) Substantial revision of the analysis is needed, particularly with regard to the criteria for sub-dividing the cloud events; 2) pertinent references prior to ca. 2000 are lacking, and would perhaps help augment the analysis.

The grammar and language in the manuscript is understandable, but could be im-

proved.

The data are both interesting and useful. However, I feel the manuscript needs substantial revision before it can be published.

Please also note the supplement to this comment:
https://www.atmos-chem-phys-discuss.net/acp-2019-660/acp-2019-660-RC2-supplement.pdf

**Supplement:**

Review of
"The evolution of cloud microphysics upon aerosol interaction at the
summit of Mt. Tai, China"
by Jiarong Li, et al.

Kevin J. Noone
Stockholm University

General Comments

The authors present very interesting measurements made at Mt. Tai in northeast China. They have an interesting set of measurements from an under-represented region. As such, it would be valuable for these data to become available. However, I feel that two factors prevent the manuscript from being published in its current form: 1) Substantial revision of the analysis is needed, particularly with regard to the criteria for sub-dividing the cloud events; 2) pertinent references prior to ca. 2000 are lacking, and would perhaps help augment the analysis.

The grammar and language in the manuscript is understandable, but could be improved.

The data are both interesting and useful. However, I feel the manuscript needs substantial revision before it can be published.

Specific Comments

P2, L10-15   The processes discussed in this paragraph have been investigated for decades, and there is a very rich literature on all the issues raised. The references here are all fairly recent – which is fine – but I feel that the addition of some citations to earlier studies would help shine a light on how rich the literature on these subjects actually is.

P2, L23   While the "first indirect effect" has become fairly accepted jargon in the cloud physics field, still it should be defined here.

P3, L5   Change "size distributions of clouds and aerosols" to "size distribution of cloud droplets and aerosol particles".

P4, entire   I feel more detail on data processing is needed. Were inversion routines used to calculate cloud droplet and aerosol particle size distributions and CCN spectra, or were these derived directly from the various instruments?

P5, L6   "claculated" should be "calculated"

P5, L9   I can't find a definition of $N_P$, which I assume is total aerosol particle number in the size range the SMPS can measure (13.6-763.5nm). Is this correct?

P5, L20-25  The comparisons to cloud conditions at in city fogs, convective and orographic clouds are interesting, but I think comparing to cloud and aerosol measurements at other mountain-top sites would be even better. There are several such sites at which various field campaigns have taken place, with fairly complete aerosol and cloud measurements. These include e.g., Mt. Kleiner Feldberg in Germany, Jungfraujoch in Switzerland, Mt. Åreskutan in Sweden, Puy-de-Dôme in France, Great Dun Fell in the U.K., Mt. Soledad in the US. Some places to start for references to data at these sites are:

**Kleiner Feldberg**: Journal of Atmospheric Chemistry 19 (1&2), 1994. Special issue on the Kleiner Feldberg Cloud Experiment 1990

**Jungfraujoch**:  Weingartner, E., et al. (2006), *Aerosol-Cloud Interactions in the Lower Free Troposphere as Measured at the High Alpine Research Station Jungfraujoch in Switzerland*.

**Åreskutan**:  Heintzenberg, J., J. A. Ogren, K. J. Noone, and L. Gärdneus (1989), The size distribution of submicrometer particles within and about stratocumulus cloud droplets on Mt. Åreskutan, Sweden, *Atmos. Res.*, *24*, 89-101.

Noone, K. J., J. A. Ogren, and J. Heintzenberg (1990), An examination of clouds at a mountain-top site in Central Sweden: the distribution of solute within cloud droplets, *Atm. Res.*, *25*, 3-15.

Targino, A. C., K. J. Noone, F. Drewnick, J. Schneider, R. Krejci, G. Olivares, S. S. Hings, and S. Borrmann (2007), Microphysical and chemical characteristics of cloud droplet residuals and interstitial particles in continental stratocumulus clouds, *Atmos. Res.*, *86*, 225-240, doi:doi:10.1016/j.atmosres.2007.05.001.

Drewnick, F., J. Schneider, S. S. Hings, N. Hock, K. J. Noone, A. Targino, S. Weimer, and S. Borrmann (2007), Measurement of Ambient, Interstitial and Residual Aerosol Particles on a Mountaintop Site in Central Sweden using an Aerosol Mass Spectrometer and a CVI, *J. Atmos. Chem.*, *56*, 1-20.

**Puy-de-Dôme**:

Asmi, E., E. Freney, H. Maxime, D. Picard, C. Rose, A. Colomb, and K. Sellegri (2012), Aerosol cloud activation in summer and winter at puy-de-Dôme high altitude site in France, *Atmospheric Chemistry and Physics Discussions*, *12*, doi:10.5194/acpd-12-23039-2012.

Laj, P., R. Dupuy, K. Sellegri, J. Pichon, J. Fournol, L. Cortes, S. Preunkert, and M. Legrand (2001), Experimental studies of aerosol- cloud droplet interactions at the puy de Dome observatory (France), *AGU Spring Meeting Abstracts.*

**Great Dun Fell**:    Choularton, T. W., et al. (1997), The Great Dun Fell Cloud Experiment 1993: An Overview, *Atmos. Environ.*, *31*(16), 2393-2405.

and the other papers in the special issue of Atmospheric Environment

**Mt. Soledad**:        Modini, R. L., et al. (2015), Primary marine aerosol-cloud interactions off the coast of California, *Journal of Geophysical Research: Atmospheres*, *120*(9), 4282-4303, doi:10.1002/2014JD022963.

Schroder, J. C., S. J. Hanna, R. L. Modini, A. L. Corrigan, S. M. Kreidenwies, A. M. Macdonald, K. J. Noone, L. M. Russell, W. R. Leaitch, and A. K. Bertram (2015), Size-resolved observations of refractory black carbon particles in cloud droplets at a marine boundary layer site, *Atmos. Chem. Phys.*, *15*(3), 1367-1383, doi:10.5194/acp-15-1367-2015.

Sanchez, K. J., et al. (2016), Meteorological and aerosol effects on marine cloud microphysical properties, *Journal of Geophysical Research: Atmospheres*, *121*(8), 4142-4161, doi:10.1002/2015JD024595.

There are more than these references, of course. These are the ones that come quickly to mind.

P6, L2-3      I find the discussion here a bit simplistic and even incorrect. LWC often tends to increase linearly with height in a cloud. If entrainment processes are active, the increase of LWC with height could still be linear, but less than the maximum adiabatic rate. Increasing $N_c$ or $r_{eff}$ does not necessarily lead to increases in LWC. You can get an increase in $N_c$ with no increase in LWC by simply having a larger number of smaller droplets. Similarly, $r_{eff}$ can increase at a constant LWC if the droplets became fewer in number but larger in size.

P6, L4        I believe "Hyderaba" should be "Hyderabad".

P6, L13       $T_a$ should be defined before it is used for the first time.

P6, L20-30 The sub-periods into which the various cloud events are divided are not clearly defined. What do "clean 1", "perturbation 2", "dissipation" and the other descriptors mean?

P6, L23       The authors divide liquid water content by cloud droplet number concentration (LWC/$N_c$) and report a value of 1.9 mg water per droplet. This is clearly erroneous. For a water density of 1 g cm$^{-3}$, this would give a droplet radius of 0.8mm, which is clearly far too large. This comment holds true for Figure 3(b) as well.

P7, L11-13   The bounding lines referred to here (and shown in Figure 3(d)) appear arbitrary. Is there any physical explanation for these lines? Why does one have a zero intercept and the other an intercept of -200?

P8, L1-13   Once again, references seem to be limited to rather recent publications. There is a wealth of literature about cloud susceptibility starting in the 1990s. I suggest starting with [Platnick, S., P. A. Durkee, K. Nielsen, J. P. Taylor, S. C. Tsay, M. D. King, R. J. Ferek, P. V. Hobbs, and J. W. Rottman (2000), The role of background cloud microphysics in the radiative formation of ship tracks, *J Atmos Sci*, *57*(16), 2607-2624] and the papers that cite this one for more comparisons.

P8, L18   I have a difficult time understanding how soluble organic particles can by hydrophobic.

P8, L28+   The discussion of Figure 5 is confusing, mostly because the figure itself is not clearly labeled. There is a great deal of information in Fig. 5; at a minimum, a clear caption is necessary. I'm afraid I can't follow the arguments presented here, and feel this material needs significant work to be understandable.

P9, L19   Given the amount of temporal variability in LWC, do hourly averages of this quantity have any real meaning?

P9, L26-27   As per my previous comments, I feel that the stages into which the authors divide cloud period 2 are arbitrary. I haven't found any explanation of these stages in terms of quantitative parameters. The physical processes discussed on pages 9-10 are certainly valid ones, and pertain to clouds in general. However, I don't find the division of the cloud events into arbitrary stages to be convincing in terms interpreting the measurements at Mt. Tai in the context of these processes. Unfortunately, I feel that Figure 6 and the discussion around it is unconvincing. There may well be interesting information here, but a clearer rationale for stratification of the data will be necessary before it can be elucidated.

P10, L27   Is there any reason to assume that the cloud thickness is 100m? My own experience measuring clouds from mountaintop sites is that cloud thickness varies quite dramatically, and at most sites is highly sensitive to changes in wind speed and direction.

Figure 2   The rightmost label in panel (e) of Figure 2 $dN/dlogD_c$, not $dN/dlogD_p$

Figure 5   This figure is very colorful, but very difficult to understand. The figure caption needs significantly more detail.

---

## Author Comment (AC1) · 26 Dec 2019

Responds to the reviewer's comments:

We sincerely thank the reviewer for the valuable comments and suggestions concerning our manuscript entitled "The evolution of cloud microphysics upon aerosol interaction at the summit of Mt. Tai, China". These comments are valuable and helpful for revising and improving our paper. The responses to reviewers are in blue. The changes are marked in red in the revised manuscript.

**Reviewer 1**

**General comments:**

This study investigates aerosol-cloud-interactions (ACI) using measurements from the high mountain site of Mt. Tai in China. As limited studies of ACI exist from high altitude measurement stations in this region, the study can potentially provide some useful data about these complex processes to the scientific community. However, the methodologies employed within this manuscript to investigate ACI are questionable, and lacking the necessary in-depth analysis currently associated with probing ACI - one of the most challenging topics currently facing the climate community. A number of conclusions presented are unsupported by the data, and rather arbitrary in nature. Numerous statements throughout the manuscript are not persuasive or lack evidence. Furthermore, the manuscript is not organised very well and the language and grammar throughout is far from the quality required for a scientific publication. Given the large concerns associated with some of the text I recommend a major revision of the entire manuscript before consideration for publication.

**Responds:** We sincerely thank you for your pertinent comments and valuable suggestions. We have revised our manuscript based on the comments from the two reviewers. We polished the language in the revised manuscript.

**Major comments:**

When investigating ACI it is of crucial importance to separate the contributions of changes in both aerosol and meteorology on any observed or simulated cloud response; as performed in other studies of ACI in the scientific literature, e.g. Malavelle et al., 2017. There does not appear to have been any attempt to account for variations in the meteorology in this study. If any of the analysis related to ACI is to remain in the manuscript, the study should include, but not limited to, the following additional analysis:

**Comment A:**

A detailed description of the measurement station with regard to location of instruments and prevailing meteorology. A picture and/or schematic is required to put the results in context of the environment in which they were measured, in particular, statistics on the height of measurements in relation to cloud base/top.

**Response:** Thank you for your comment. We add two graphs (Fig. S1 and Fig. S2) and detailed descriptions of location of instruments and prevailing meteorology in section 2.1 of the revised manuscript (Page 3 Line 28 to Page 4 Line 6):

"From 17 June to 30 July 2018, 40 cloud events in total were monitored at the Shandong Taishan

Meteorological Station at summit of Mt. Tai (Tai'an, China; 117°13' E, 36°18' N; 1545 m a.s.l.; Fig. S1).

Mt. Tai is the highest point in the central of North China Plain (NCP) and located within the transportation channel between the NCP and the Yangtze River Delta (Shen et al., 2019). The altitude of Mt. Tai is close to 1.6 km, which is close to the top of the planetary boundary layer in Central East China and usually sited for the characteristic of particles inputting to clouds (Hudson, 2007). Local cloud events frequently occurred at the summit of Mt. Tai, especially in summer. As shown in Fig. S2, the prevailing wind direction during this summer campaign is east wind (23.3%), southwest wind (22.8%) and south wind (21.9%), respectively. About 85.6% of wind speed was less than 8 m s-1. While the monitored cloud events in the present study was mainly influence by south wind (34.7%) and southwest wind (22%). The arrangement of instruments was presented in Fig. S1(b)."

Figure S1. The pictures and schematic of (a) the measurement station (printscreen from Google Map) (b) the arrangement of instruments in Shandong Taishan Meteorological Station (http://p.weather.com.cn/2016/12/2638460.shtml). The corresponding sampling tubes were at least 1.5 m higher than the roof and at least 1.0 m away from each other to avoid the mutual interference.

Figure S2. Wind direction and wind speed a) during the whole summer campaign at Mt. Tai, b) without cloud events and c) during cloud events.

Due to the lack of corresponding instruments, we cannot directly get the information of cloud base height (CBH) and cloud top height. Based on the meteorological data on the ground level,

the lifting condensation level (LCL) was calculated and applied to approximate CBH as shown in Section 2.6. We added the information of CBH in Fig. 2b (Page 6 Line 12 - 23):

**"2.6. Calculation of cloud base height**

In the present study, the estimated lifting condensation level (LCL) is applied to represent the cloud base height (CBH) due to the lack of corresponding instruments. The calculation of LCL depends on the meteorological parameters measured at Tai'an Station. The ground-level data of temperature, dew point temperature, and pressure were used as input parameters (Georgakakos and Bras, 1984):

$$p_{LCL} = \frac{1}{(\frac{T_g - T_{gd}}{223.15} + 1)^{3.5}} \times p_g$$
$$T_{LCL} = \frac{1}{(\frac{T_g - T_{gd}}{223.15} + 1)} \times T_g$$
$$CBH = 18400 \times (1 + \frac{T_{LCL} - T_g}{273}) \times \lg \frac{p_g}{p_{LCL}}$$

Where  $p_{LCL}$  is the LCL pressure;  $T_{LCL}$  is the LCL temperature.

During the observation period, CBH ranged from 460.3 m to 3639.1 m with the average value of 1382.5 m. As shown in Fig. 2b, the observation station would be totally enveloped in clouds and around when cloud events occurred. The corresponding distance between the observation point and CBH was represented in Fig. 2b."

---

## Author Comment (AC2) · 26 Dec 2019

Responds to the reviewer's comments:

We sincerely thank the reviewer for the valuable comments and suggestions concerning our manuscript entitled "The evolution of cloud microphysics upon aerosol interaction at the summit of Mt. Tai, China". These comments are all valuable and helpful for revising and improving our paper. The responses to reviewers are in blue. The changes are marked in red in the revised manuscript. The Tables and Figures of the revised manuscript were presented at the end of the Responds.

**Reviewer 2**

**General Comments**

**Comment 1:**

The authors present very interesting measurements made at Mt. Tai in northeast China. They have an interesting set of measurements from an under-represented region. As such, it would be valuable for these data to become available. However, I feel that two factors prevent the manuscript from being published in its current form: 1) Substantial revision of the analysis is needed, particularly with regard to the criteria for sub-dividing the cloud events; 2) pertinent references prior to ca. 2000 are lacking, and would perhaps help augment the analysis.

**Response:** Thank you for your valuable comments. We added descriptions about the criteria for sub-dividing the cloud events in Section 3.2 (Page 8 Line 21 to Page 9 Line 8).

"CP-1 was separated into four stages, including SC1 (stage-clean 1), SP1 (stage-perturbation 1), SC2 (stage-clean 2), and SP2 (stage-perturbation 2) based on whether the perturbation of particles occurred (Fig. 3b). The characteristics of SC1 and SC2 were low $N_C$ (383 # cm$^{-3}$ and 347 # cm$^{-3}$, respectively), large $r_{eff}$ (7.26 μm and 6.36 μm, respectively) and high LWC/$N_C$ (1.01 ng #$^{-1}$ and 0.75 ng #$^{-1}$, respectively, which represents averaged water each cloud droplet contained) (Fig. 3b). During SP1 and SP2, the perturbation through particles occurred. Dramatic increase of $N_C$ (949 # cm$^{-3}$ and 847 # cm$^{-3}$, respectively) and decrease of $r_{eff}$ (4.90 μm and 4.88 μm, respectively) and LWC/$N_C$ (0.35 ng #$^{-1}$ and 0.36 ng #$^{-1}$, respectively) was caused.

Each cloud event of CP-2 was separated into activation stage (S1), collision-coalescence stage (S2), stable stage (S3), and dissipation stage (S4) according to the regular changes of $N_C$ and LWC/$N_C$ (Fig. 3a). In S1, $N_C$ dramatically increased to its maximum value among the cloud events. In S2, $N_C$ declined sharply to a stable value, meanwhile LWC/$N_C$ reached the maximum value. In S3, $N_C$ was stable or slightly varied and LWC/$N_C$ started to decrease. In S4, both $N_C$ and LWC/$N_C$ decreased sharply again and finally arrived zero. Even though the two stages (S2 and S3) in CE-25 were not totally follow the division rules, the other six cloud events followed well. It indicated that the division was helpful to study the variations of cloud microphysical properties during CP-2. The newly formed cloud droplets during

S1 were characterized by small size, high $N_C$ and low $LWC/N_C$ values (Fig. 2f and 3b). For example, about 2310 # $cm^{-3}$ of cloud droplets can quickly form in the first 2 hours of CE-20. The $r_{eff}$ of these droplets was smaller than 4.1 μm and $LWC/N_C$ was about 0.2 ng $\#^{-1}$. In going from S2 to S3, the strong collision-coalescence between cloud droplets caused the increase of both $r_{eff}$ and $LWC/N_C$. In S4, the increase of $PM_{2.5}$, $N_P$ and $T_a$ (Fig. 2b and Fig. 2c) decreased cloud droplet sizes (Rosenfeld et al., 2014a), decreased the ambient supersaturation, enhanced the evaporation of small droplets (Ackerman et al., 2004), and finally caused the vanishment of cloud events (Mazoyer et al., 2019)."

We cited many useful researches prior to ca. 2000 in the revised manuscript to help us improve our analysis. Such as

Page 2 Line 10 - 16:

"The cloud processes can incorporate large amount of fine particulate mass (Heintzenberg et al., 1989), change the size distributions (Drewnick et al., 2007;Schroder et al., 2015) and alter the CCN compositions through homogeneous and heterogeneous reactions (Roth et al., 2016). In addition, the variation of aerosol number concentrations and size distributions could alter the cloud microphysics. Through studying microphysical characteristics of cloud droplet residuals at Mt. Åreskutan, Noone et al. (1990) found that larger cloud droplets preferred to form on larger Cloud Condensation Nuclei (CCN)."

Page 8 Line 2 - 3:

"Different from convective clouds studied by research aircraft, orographic clouds were mainly formed in the boundary layer as air approaching the ridge, forced to rise up and cooled by adiabatic expansion (Choularton et al., 1997)."

Page 10 Line 24 – 28:

"What's more, the perturbation of aerosol particles would cause stronger albedo enhancements when pollution is low in the ambient air (Platnick et al., 2000). Through studying the impact of ship-produced aerosols on the microstructure and albedo of warm marine stratocumulus clouds, Durkee et al. (2000) found that the clean and shallow boundary layers would be more readily perturbed by the addition of ship particle effluents."

**Comment 2:**

The grammar and language in the manuscript is understandable, but could be improved.

**Response:** We improved our grammar and language in the revised manuscript.

**Comment 3:**

The data are both interesting and useful. However, I feel the manuscript needs

substantial revision before it can be published.

**Response:** We carefully revised our manuscript based on the comments from the reviewers.

**Specific Comments**

**Comment 1:**

P2, L10-15 The processes discussed in this paragraph have been investigated for decades, and there is a very rich literature on all the issues raised. The references here are all fairly recent – which is fine – but I feel that the addition of some citations to earlier studies would help shine a light on how rich the literature on these subjects actually is.

**Response:** We sincerely thank you for your pertinent comments and valuable suggestions. We cited some valuable papers published before 2000. Such as

Page 2 Line 10 - 16:

"The cloud processes can incorporate large amount of fine particulate mass (Heintzenberg et al., 1989), change the size distributions (Drewnick et al., 2007;Schroder et al., 2015) and alter the CCN compositions through homogeneous and heterogeneous reactions (Roth et al., 2016). In addition, the variation of aerosol number concentrations and size distributions could alter the cloud microphysics. Through studying microphysical characteristics of cloud droplet residuals at Mt. Åreskutan, Noone et al. (1990) found that larger cloud droplets preferred to form on larger Cloud Condensation Nuclei (CCN)."

Page 8 Line 2 - 3:

"Different from convective clouds studied by research aircraft, orographic clouds were mainly formed in the boundary layer as air approaching the ridge, forced to rise up and cooled by adiabatic expansion (Choularton et al., 1997)."

Page 10 Line 24 – 28:

"What's more, the perturbation of aerosol particles would cause stronger albedo enhancements when pollution is low in the ambient air (Platnick et al., 2000). Through studying the impact of ship-produced aerosols on the microstructure and albedo of warm marine stratocumulus clouds, Durkee et al. (2000) found that the clean and shallow boundary layers would be more readily perturbed by the addition of ship particle effluents."

**Comment 2:**

P2, L23 While the "first indirect effect" has become fairly accepted jargon in the cloud physics field, still it should be defined here.

**Response:** We add the definition of "first indirect effect" in the revised manuscript (Page 1 Line 21 – 22).

"For a given liquid water content, aerosol particles can act as CCN, lead to higher number concentrations

of cloud droplets with smaller sizes and result in higher albedo (Twomey effect or first indirect effect, FIE) (Twomey, 1974)."

**Comment 3:**

P3, L5 Change "size distributions of clouds and aerosols" to "size distribution of cloud droplets and aerosol particles".

**Response:** Thank you. We have revised it in the revised manuscript (Page 3 Line 8 - 9).

"However, lacking knowledge of the size distributions of cloud droplets and aerosol particles makes it difficult to evaluate the cloud microphysics in small-scale regions (Fan et al., 2016;Khain et al., 2015;Sant et al., 2013)"

**Comment 4:**

P4, entire I feel more detail on data processing is needed. Were inversion routines used to calculate cloud droplet and aerosol particle size distributions and CCN spectra, or were these derived directly from the various instruments?

**Response:** We added more detailed information about the calibrations of instruments and the corrections of the data in Section 2.2 and Section 2.3 (Page 4 Line 7 to Page 5 Line 28). The CCN spectra was derived directly from the calibrated CCN counter.

"**2.2 Cloud microphysical parameters**

[revised manuscript text omitted]

**Comment 5:**
P5, L6 "claculated" should be "calculated"
**Response:** Thank you. We have revised it in the revised manuscript (Page 6 Line 26).

"In the present study, FIE based either on the $r_{eff}$ or on $N_C$ were used calculated as"

**Comment 6:**
P5, L9 I can't find a definition of $N_P$, which I assume is total aerosol particle number in the size range the SMPS can measure (13.6-763.5nm). Is this correct?
**Response:** Yes. We added the definition of $N_P$ in the revised manuscript (Page 5 Line 26 - 28).

"In the present study, $PM_{2.5}$ and $N_P$ (the total number concentration of aerosol particles measured by SMPS) were combined together to separate aerosol conditions of cloud processes."

**Comment 7:**

P5, L20-25 The comparisons to cloud conditions at in city fogs, convective and orographic clouds are interesting, but I think comparing to cloud and aerosol measurements at other mountain-top sites would be even better. There are several such sites at which various field campaigns have taken place, with fairly complete aerosol and cloud measurements. These include e.g., Mt. Kleiner Feldberg in Germany, Jungfraujoch in Switzerland, Mt. Åreskutan in Sweden, Puy-de-Dôme in France, Great Dun Fell in the U.K., Mt. Soledad in the US. Some places to start for references to data at these sites are: (recommend papers)

**Response:** Thank you for your comment. In this part, we want to give an overview of the ranges of the monitored cloud/fog microphysical properties such as $N_C$, LWC and $r_{eff}$/MVD. Even though we did not find the corresponding ranges in the the suggested papers, these papers gave abundant observation studies involving size distributions of aerosols and cloud droplets, microphysical and chemical characteristics of cloud droplet residuals/interstitial particles and meteorological and aerosol effects on clouds. We cited them in the revised manuscriptto help us comprehensively discuss the aerosol-cloud interactions at Mt. Tai. Such as

Page 2 Line 10 - 16:

"The cloud processes can incorporate large amount of fine particulate mass (Heintzenberg et al., 1989), change the size distributions (Drewnick et al., 2007;Schroder et al., 2015) and alter the CCN compositions through homogeneous and heterogeneous reactions (Roth et al., 2016). In addition, the variation of aerosol number concentrations and size distributions could alter the cloud microphysics. Through studying microphysical characteristics of cloud droplet residuals at Mt. Åreskutan, Noone et al. (1990) found that larger cloud droplets preferred to form on larger Cloud Condensation Nuclei (CCN)."

Page 8 Line 2 - 3:

"Different from convective clouds studied by research aircraft, orographic clouds were mainly formed in the boundary layer as air approaching the ridge, forced to rise up and cooled by adiabatic expansion (Choularton et al., 1997)."

Page 9 Line 14 - 15:

"In contrast, Modini et al. (2015) found negative relation between $N_C$ and the number of particles with diameters larger than 100 nm due to the reduction of supersaturation by coarse primary marine aerosol particles."

Page 9 Line 30 to Page 10 Line 2:

"In the study of Mazoyer et al. (2019) and Asmi et al. (2012), both of them found that high $N_{CCN}/N_P$ was associated with high $\kappa$ at a given SS. Thus, $N_{CCN,0.2}$ ($N_{CCN}$ measured at SS = 0.2%) to $N_P$ fractions ($N_{CCN,0.2}/N_P$, CCN activation ratio) is applied to reflect the hygroscopicity of ambient aerosols at Mt. Tai."

Page 12 Line 3 - 6:

"During CP-2, aerosol particles with diameters larger than 150 nm quickly decreased by activation when

cloud events occurred, while the number of aerosol particles in the size of 50-150 nm were slightly influenced by cloud events (the first panel of Fig. 5a). It was consistent with the study of Targino et al. (2007) who found aerosol size distributions of cloud residuals, which represented aerosol particles activated to cloud droplets, peaked at about 0.15 μm at Mt. Åreskutan."

**Comment 8:**
P6, L2-3 I find the discussion here a bit simplistic and even incorrect. LWC often tends to increase linearly with height in a cloud. If entrainment processes are active, the increase of LWC with height could still be linear, but less than the maximum adiabatic rate. Increasing $N_c$ or $r_{eff}$ does not necessarily lead to increases in LWC. You can get an increase in $N_c$ with no increase in LWC by simply having a larger number of smaller droplets. Similarly, $r_{eff}$ can increase at a constant LWC if the droplets became fewer in number but larger in size.

Response: Thank you for your comment. Increasing $N_c$ or $r_{eff}$ does not necessarily lead to increases in LWC. But if LWC increased, it should be influenced by the increase of one of $r_{eff}$ and $N_C$ or both of them. We changed the expression as shown in the revised manuscript (Page 7 Line 24 - 25).

"The increase of LWC should be determined by the increase of $r_{eff}$ and/or $N_C$."

**Comment 9:**
P6, L4 I believe "Hyderaba" should be "Hyderabad".
Response: Yes. Thank you for your comment. We corrected it in the revised manuscript.

**Comment 10:**
P6, L13 $T_a$ should be defined before it is used for the first time
Response: We added the definition of $T_a$ in Section 2.5 (Page 6 Line 8 – 10).

"Meteorological parameters including the ambient temperature ($T_a$, ℃), relative humidity (RH), wind speed (WS, m s$^{-1}$) and wind direction (WD, ⁹) were provided by Shandong Taishan Meteorological Station at the same observation point."

**Comment 11:**
P6, L20-30 The sub-periods into which the various cloud events are divided are not clearly defined. What do "clean 1", "perturbation 2", "dissipation" and the other descriptors mean?
Response: Based on whether the perturbation of particles occurred, CP-1 was divided into four stages. Compared with two clean stages (SC1 and SC2), two stages with perturbation of aerosol particles (SP1 and SP2) were characterized with higher $N_C$, smaller $r_{eff}$ and lower LWC/$N_C$. The averaged characteristic values of $N_C$, $r_{eff}$ and LWC/$N_C$ during SP1, SP2, SC1 and SC2 were

added in the revised manuscript. According to the regular changes of $N_C$ and LWC/$N_C$, each cloud event of CP-2 was divided into four stages. The variation of $N_C$ during the four stages was described in the Page 8 Line 21 to Page 9 Line 8.

"CP-1 was separated into four stages, including SC1 (stage-clean 1), SP1 (stage-perturbation 1), SC2 (stage-clean 2), and SP2 (stage-perturbation 2) based on whether the perturbation of particles occurred (Fig. 3b). The characteristics of SC1 and SC2 were low $N_C$ (383 # cm$^{-3}$ and 347 # cm$^{-3}$, respectively), large $r_{eff}$ (7.26 μm and 6.36 μm, respectively) and high LWC/$N_C$ (1.01 ng #$^{-1}$ and 0.75 ng #$^{-1}$, respectively, which represents averaged water each cloud droplet contained) (Fig. 3b). During SP1 and SP2, the perturbation through particles occurred. Dramatic increase of $N_C$ (949 # cm$^{-3}$ and 847 # cm$^{-3}$, respectively) and decrease of $r_{eff}$ (4.90 μm and 4.88 μm, respectively) and LWC/$N_C$ (0.35 ng #$^{-1}$ and 0.36 ng #$^{-1}$, respectively) was caused.

Each cloud event of CP-2 was separated into activation stage (S1), collision-coalescence stage (S2), stable stage (S3), and dissipation stage (S4) according to the regular changes of $N_C$ and LWC/$N_C$ (Fig. 3a). In S1, $N_C$ dramatically increased to its maximum value among the cloud events. In S2, $N_C$ declined sharply to a stable value, meanwhile LWC/$N_C$ reached the maximum value. In S3, $N_C$ was stable or slightly varied and LWC/$N_C$ started to decrease. In S4, both $N_C$ and LWC/ $N_C$ decreased sharply again and finally arrived zero. Even though the two stages (S2 and S3) in CE-25 were not totally follow the division rules, the other six cloud events followed well. It indicated that the division was helpful to study the variations of cloud microphysical properties during CP-2. The newly formed cloud droplets during S1 were characterized by small size, high $N_C$ and low LWC/$N_C$ values (Fig. 2f and 3b). For example, about 2310 # cm$^{-3}$ of cloud droplets can quickly form in the first 2 hours of CE-20. The $r_{eff}$ of these droplets was smaller than 4.1 μm and LWC/$N_C$ was about 0.2 ng #$^{-1}$. In going from S2 to S3, the strong collision-coalescence between cloud droplets caused the increase of both $r_{eff}$ and LWC/$N_C$. In S4, the increase of PM$_{2.5}$, $N_P$ and $T_a$ (Fig. 2b and Fig. 2c) decreased cloud droplet sizes (Rosenfeld et al., 2014a), decreased the ambient supersaturation, enhanced the evaporation of small droplets (Ackerman et al., 2004), and finally caused the vanishment of cloud events (Mazoyer et al., 2019)."

**Comment 12:**
P6, L23 The authors divide liquid water content by cloud droplet number concentration (LWC/$N_c$) and report a value of 1.9 mg water per droplet. This is clearly erroneous. For a water density of 1 g cm$_{-3}$, this would give a droplet radius of 0.8mm, which is clearly far too large. This comment holds true for Figure 3(b) as well.

**Response:** Thank you for your comment. We checked our data and found that we misused the unit of $N_C$ when we calculated the values of $LWC/N_C$. The unit of $N_C$ should be # $cm^{-3}$ instead of # $m^{-3}$. Thus, the result was overestimated with a factor of $10^6$. The number is correct but the unit should be ng $\#^{-1}$ for $LWC/N_C$. We have corrected the corresponding units in the text and in Fig. 3(b). (Page 8 Line 22 - 26)

"The characteristics of SC1 and SC2 were low $N_C$ (383 # $cm^{-3}$ and 347 # $cm^{-3}$, respectively), large $r_{eff}$ (7.26 μm and 6.36 μm, respectively) and high $LWC/N_C$ (1.01 ng $\#^{-1}$ and 0.75 ng $\#^{-1}$, respectively, which represents averaged water each cloud droplet contained) (Fig. 3b). During SP1 and SP2, the perturbation through particles occurred. Dramatic increase of $N_C$ (949 # $cm^{-3}$ and 847 # $cm^{-3}$, respectively) and decrease of $r_{eff}$ (4.90 μm and 4.88 μm, respectively) and $LWC/N_C$ (0.35 ng $\#^{-1}$ and 0.36 ng $\#^{-1}$, respectively) was caused."

**Comment 13:**
P7, L11-13 The bounding lines referred to here (and shown in Figure 3(d)) appear arbitrary. Is there any physical explanation for these lines? Why does one have a zero intercept and the other an intercept of -200?

**Response:** Due to the limitation of instruments, we could not directly get the hygroscopicity parameter $\kappa$. In the study of Asmi et al. (2012), they conducted the study at Puy-de-Dome and discussed the relations between the number concentration of CCN, the number concentration of aerosol particles and the hygroscopicity parameter $\kappa$. The two dashed linear lines represented the visually defined boundaries in within most of the data at Puy-de-Dome are centered. We cited these two lines and wanted to compare their data with ours. We found that most of our data was also centered between these two dashed lines. Asmi et al. (2012) found that a good linear fit of CCN versus $N_P$ and higher values of $\kappa$ existed in winter. The data of CP-2 (Figure 3d) at Mt. Tai was similar with the winter data at Puy-de-Dome that good linear fits of CCN versus $N_P$ existed. Thus, we speculated that the values of $\kappa$ of CP-2 might be higher than that of CP-1. We rephrased the corresponding part in the revised manuscript in Page 9 Line 29 to Page 10 Line 10.

"The hygroscopicity of aerosols determines the ability of aerosols acted as CCN, which can further influence cloud number concentrations. Due to the lack of corresponding instruments, the hygroscopicity parameter $\kappa$ is not available. In the study of Mazoyer et al. (2019) and Asmi et al. (2012), both of them found that high $N_{CCN}/N_P$ was associated with high $\kappa$ at a given SS. Thus, $N_{CCN,0.2}$ ($N_{CCN}$ measured at SS = 0.2%) to $N_P$ fractions ($N_{CCN,0.2}/N_P$, CCN activation ratio) is applied to reflect the hygroscopicity of ambient aerosols at Mt. Tai. As shown in Fig. 3b $N_{CCN,0.2}/N_P$ ranged from 0.06 to 0.69 in CP-1 yet it was range from 0.22 to 0.66 in CP-2. The plot of $N_{CCN,0.2}$ versus $N_P$ was more scatter in CP-1 than that in CP-2 (Fig. 3b and Fig. 3c). Values lower than 0.22 did not appear during CP-2. Even though the settled SS

in the present study (SS = 0.2%) is different from that at puy-de-Dome (SS = 0.24%), most of the data points of CP-1 and CP-2 were distributed between the two recommended dashed lines (the visually defined boundaries in within most of the data are centered, Fig. 3c and 3d) by Asmi et al. (2012). During the observation program at Puy-de-Dome, France, Asmi et al. (2012) found that higher $N_{CCN}/N_P$ and more concentrated plot of $N_{CCN,0.2}$ versus $N_P$ were usually occurred during winter when higher fraction of aged organics was observed. It indicated that the difference of aerosol organic chemical compositions during CP-1 and CP-2 might influence the $\kappa$ of aerosols and further affect the $N_{CCN}/N_P$ ratio during this two cloud processes."

**Comment 14:**

P8, L1-13 Once again, references seem to be limited to rather recent publications. There is a wealth of literature about cloud susceptibility starting in the 1990s. I suggest starting with [Platnick, S., P. A. Durkee, K. Nielsen, J. P. Taylor, S. C. Tsay, M. D. King, R. J. Ferek, P. V. Hobbs, and J. W. Rottman (2000), The role of background cloud microphysics in the radiative formation of ship tracks, *J Atmos Sci*, *57*(16), 2607-2624] and the papers that cite this one for more comparisons.

**Response:** Thank you for your suggestions. The papers recommend by the reviewer give information on the sensitivity of clouds to changes in aerosol particles. They are helpful to augment discussion in Section 3.2.2. We cited the studies of Platnick et al., (2000) and Durkee et al. (2000) in Section 3.2.2 (Page 10 Line 24 - 28).

"What's more, the perturbation of aerosol particles would cause stronger albedo enhancements when pollution is low in the ambient air (Platnick et al., 2000). Through studying the impact of ship-produced aerosols on the microstructure and albedo of warm marine stratocumulus clouds, Durkee et al. (2000) found that the clean and shallow boundary layers would be more readily perturbed by the addition of ship particle effluents."

**Comment 15:**

P8, L18 I have a difficult time understanding how soluble organic particles can by hydrophobic.

**Response:** Thank you for your careful review. The SSO should be "slightly soluble organics" Yuan et al., (2008). We made corrections in the revised manuscript. (Page 11 Line 10 - 13)

"By using the 2-D Goddard Cumulus Ensemble model (GCE), Yuan et al. (2008) explained that the positive relationship between $r_{eff}$ and AOD appeared to originate from the increasing slightly soluble organics (SSO) particles. The increase of SSO would act to increase of the critical supersaturation for

particles to be activated and resulted in less numbers of activated particles."

**Comment 16:**

P8, L28+ The discussion of Figure 5 is confusing, mostly because the figure itself is not clearly labeled. There is a great deal of information in Fig. 5; at a minimum, a clear caption is necessary. I'm afraid I can't follow the arguments presented here, and feel this material needs significant work to be understandable.

**Response:** We added detailed captions in the revised manuscript. (Page 11 Line 22 to Page 12 Line 9)

"**3.2.3 Size distribution of cloud droplets and particles**

To illustrate the evolution of the aerosol particles and the cloud droplets during the cloud processes, the size distributions of $N_P$ and $N_C$ during different cloud stages are plotted in Fig. 5. For each of the four size bins ranged from 2 to 13 μm, cloud number concentrations of SC1 and SC2 were lower than those of SP1 and SP2. In the size bin of 13–50 μm, however, $N_C$ of SC1 and SC2 were the largest (Fig. 5b). This size distributions of cloud droplets in SC1 and SC2 resulted in the larger $r_{eff}$ during the two stages, which was consistent with the result shown in Fig. 3b. During two perturbation stages of SP1 and SP2 in CP-1, the numbers of aerosol particles in all size bins increased. But the increase of aerosol particles larger than 150 nm was the smallest, indicating that aerosols larger than 150 nm were more easily activated into cloud droplets. The activation of aerosol particles with the size larger than 150 nm in the present study dramatically increased $N_C$ of 5–10 μm and made $N_C$ of SP1 and SP2 in different size bins all comparable with those of CP-2 (Fig. 5b).

As shown in Fig. 5c, cloud droplets with $D_C$ ranging from 5 to 10 μm had high $N_C$ in each stage in CP-2 and cloud droplets with $D_C$ ranging from 13 to 50 μm had low $N_C$ in each stage if compared to CP-1. It caused the lower $r_{eff}$ in CP-2 than CP-1. During CP-2, aerosol particles with diameters larger than 150 nm quickly decreased by activation when cloud events occurred, while the number of aerosol particles in the size of 50-150 nm were slightly influenced by cloud events (the first panel of Fig. 5a). It was consistent with the study of Targino et al. (2007) who found aerosol size distributions of cloud residuals, which represented aerosol particles activated to cloud droplets, peaked at about 0.15 μm at Mt. Åreskutan. Mertes et al. (2005) also found that particles centered at $d_P$ = 200 nm could be efficiently activated to droplets while most Aitken mode particles remained in the interstitial phase. Compared with other stages, S1 had the highest $N_C$ in three size bins of [2, 5) μm and [5, 7) μm. It indicated that large numbers of cloud droplets with small sizes were formed in the beginning of cloud events in CP-2."

**Comment 17:**

P9, L19 Given the amount of temporal variability in LWC, do hourly averages of this quantity have any real meaning?

**Response:** The time resolution of the corresponding data in Figure 6(a) should be 5 min. We have corrected it in the revised manuscript (Page 12 Line 11 – 12). As shown in Figure R1, the relations between LWC and $r_{eff}$ were consistent even though data with different time resolutions (1 min and 5 min) were applied. In order to make the picture clearer, we choose the 5 min averaged data to plot Figure 6(a). However, the data in Figure 6(c) was 50 min averaged, which was depended on the time resolution of CCN. We added the description of time resolutions we applied in the figure caption.

[Figure]

Figure R1. The plot of LWC versus $r_{eff}$ of CP-1 and CP-2. Time resolutions of the corresponding data were 5 min and 1 min, respectively.

"The 5 min averaged LWC for CP-1 and CP-2 is plotted against corresponding $r_{eff}$ in Fig. 6a. Large cloud droplets ($r_{eff} > 8$ μm) were observed in CP-1, while the $r_{eff}$ for CP-2 varied narrowly in the range of 2.5– 8 μm."

**Comment 18:**

P9, L26-27 As per my previous comments, I feel that the stages into which the authors divide cloud period 2 are arbitrary. I haven't found any explanation of these stages in terms of quantitative parameters. The physical processes discussed on pages 9-10 are certainly valid ones, and pertain to clouds in general. However, I don't find the division of the cloud events into arbitrary stages to be convincing in terms interpreting the measurements at Mt. Tai in the context of these processes. Unfortunately, I feel that Figure 6 and the discussion around it is unconvincing. There may well be interesting information here, but a clearer rationale for stratification of the data will be necessary before it can be elucidated.

**Response:** During each cloud event of CP-2, the variations of $N_C$ and LWC/$N_C$ were applied to divide different stages of cloud events. The corresponding rules were detailedly described in Page 8 Line 27 to Page 9 Line 8.

[revised manuscript text omitted]

**Comment 19:**
P10, L27 Is there any reason to assume that the cloud thickness is 100m? My own experience measuring clouds from mountaintop sites is that cloud thickness varies quite dramatically, and at most sites is highly sensitive to changes in wind speed and direction.

**Response:** Based on the equations in Section 2.8, albedo depends on the values of LWC, $N_C$ and cloud thickness. Here, we set the same cloud thickness for CP-1 and CP-2, and discuss the difference between albedo due to the change of LWC and $N_C$. Unfortunately, we don't have the corresponding data of cloud thickness during our monitoring program. In the revised manuscript, we applied the averaged values of LWC and $N_C$ of CP-1 and CP-2 to calculate the corresponding albedo during CP-1 and CP-2. For a given cloud thickness, albedo during CP-2 was always higher than that during CP-1 if the cloud thickness is lower than about 2500 m (Fig. 6d). We revised this part in Page 13 Line 19 - Line 24.

"The thickness of orographic cloud was usually very thin (Welch et al., 2008). If assuming the cloud thickness during CP-1 and CP-2 were equal, albedo would depend on the values of LWC and $N_C$ as described in Section 2.8. Cloud albedo during CP-2 was always higher than that during CP-1, especially when the cloud thickness was lower than about 2500 m (Fig. 6d). Through studying marine stratocumulus clouds in the north-eastern Pacific Ocean, Twohy et al. (2005) also found that the increase of $N_C$ by a factor of 2.8 would lead to 40% increase of albedo going from 0.325 to 0.458. It indicated that the higher $N_C$ would increase the cloud albedo if assuming no change of cloud thickness."

**Comment 20:**

Figure 2 The rightmost label in panel (e) of Figure 2 dN/dlogD$_c$, not dN/dlogD$_p$

**Response:** Thank you for your comment. We have corrected the label in the revised manuscript.

**Comment 21:**

Figure 5 This figure is very colorful, but very difficult to understand. The figure caption needs significantly more detail.

**Response:** We added detailed captions in the revised manuscript.

**Tables and Figures**

[revised manuscript text omitted]

[a]The value of $\partial lnN_C/\partial lnN_P$ was equal to $FIE_N$

[b]$R^2$ represented correlation coefficient

[Figure]

**Figure S1. The pictures and schematic of (a) the measurement station (printscreen from Google Map) (b) the arrangement of instruments in Shandong Taishan Meteorological Station (http://p.weather.com.cn/2016/12/2638460.shtml). The corresponding sampling tubes were at least 1.5 m higher than the roof and at least 1.0 m away from each other to avoid the mutual interference.**

[Figure]

**Figure S2. Wind direction and wind speed a) during the whole summer campaign at Mt. Tai, b) without cloud events and c) during cloud events.**

[Figure]

**Figure S3. Influence of the topography on the vertical wind field at monitoring station. Taking (a) the south-north transect of Mt. Tai and (b) the southwest-northeast transect of Mt. Tai to estimate the inclination angles and updraft velocities.**

[Figure]

**Figure S4. The averaged inorganic chemical compositions of cloud samples collected during CP-1 and CP-2. Each cloud process contained 12 cloud samples.**

[revised manuscript text omitted]

---

## Author Response (AR2)

**Responses to the comments from Anonymous Referee #3:**

Comments on "The evolution of cloud microphysics upon aerosol interaction at the summit of Mt. Tai, China" by Li et al.

The authors have greatly improved the quality of this paper compared to the last version. I would recommend its publication if the authors could address my following concerns.

Compared to results in earlier studies is an important approach to generalize your finding, or find new questions. Thus, such comparison should end up with concluding remarks. I would suggest the authors to revise the manuscript accordingly. Here are a few examples:

We sincerely thank the reviewer for the positive comment on our revised manuscript, and for the valuable comments and suggestions. In the following, we have addressed the reviewers' comments one by one. Comments by the reviewers are given in black normal font, and our response to the comments is shown in blue. Newly added and modified text in the revised manuscript and supporting material is given in red.

**Comment 1:**

Page 7 line 21 "The number concentration of cloud droplets at Mt. Tai both in the present study and in 2014 can reach 2000-3000 # cm-3 (Li et al., 2017a), which is much higher than those values (with a range 25 of 10–700 # cm-3) for city fogs and convective and orographic clouds", so?

**Response:** We sincerely thank you for your pertinent comments and valuable suggestions. We added the concluding remark at the end of this paragraph.

Page 7 Line 8-9: "It represented clouds at Mt. Tai were characterized with high NC."

**Comment 2:**

Page 7 line 27, "When compared with previous orographic clouds, LWC at Mt. Tai appeared to show a larger range. We monitored the high values, which are comparable with convective clouds, and the low values, which are similar to city fogs." This seems to be the conclusion of this paragraph, so?

**Response:** We added the concluding remarks. We rewrote this paragraph to make it more understandable. **Page 7 Line 10-21:** "The microphysics of different clouds and fogs can generally be distinguished in a plot of  $r_{eff}$  (or *MVD*) against *LWC*. As illustrated in Fig. 1, the *LWC* increases as the altitude increases generally in order of city fogs, orographic clouds and convective clouds, and Mt. Tai generally according the rule. It is consistent with the study by Penner et al. (2004) that *LWC* within clouds increases linearly with altitude. For *LWC* values of clouds at Mt. Tai, we monitored the high values, which are comparable with convective clouds, and the low values, which are similar to city fogs (Fig. 1). It indicated that clouds at Mt. Tai appeared to show a larger range of *LWC* values. The increase of *LWC* at Mt. Tai should be determined by the increase of  $r_{eff}$  and LWC in cloud event 20 (CE-20) were 1519 # cm-3, 5.2 µm and 0.54 g m-3, respectively, while the corresponding values in CE-16 were 59 # cm-3, 9.8 µm and 0.14 g m-3, respectively. Even though  $r_{eff}$  of CE-20 was smaller compared with CE-16, but the higher  $N_C$  determined the larger *LWC* of clouds in CE-20. In the following parts, the evolution of cloud and aerosol microphysical properties were presented. The influence of meteorological parameters (such as updraft velocity and cloud base height) and aerosol particle on cloud microphysics were discussed."

**Comment 3:**

Figure 1, can you explain the meaning of lines? "The dashed and solid shapes indicated the airborne and land observation, respectively." What's the meaning of the position, area of these rectangles?

**Response:** The rectangle areas in Fig. 1 represented the range of data obtained from airborne and land observations. We rewrite the caption of Fig. 1 in the manuscript in **Page 22**. Through presented the range of the LWC and the size of cloud/fog droplets. We found that 1) the LWC increases as the altitude increases generally in order of city fogs, orographic clouds and convective clouds; 2) clouds at Mt. Tai appeared to show a larger range of LWC values.

---

## Author Response (AR3)

The authors have made most corrections, and there is one comment not adequately addressed.

Comment 5: Page 9 line 5, "the increase of PM2.5, Np and Ta decreased cloud droplet sizes(Rosenfeld et al., 2014a), decreased the ambient supersaturation, enhanced the evaporation of small droplets (Ackerman et al., 2004), and finally caused the cloud events to vanish (Mazoyer et al., 2019)." Here the authors claimed that aerosols enhanced evaporation and caused cloud events to vanish by referring to early studies. This statement, however, is in contrast to the authors' statement in the abstract "we find that the albedo can increase 36.4% if the cloud gets to be disturbed by aerosols. This may induce a cooling effect on the local climate system". Has this lifetime effect been considered in your calculation?

Enhanced evaporation will reduce the lifetime and cloud, resulting in a warming effect. Since you made this statement and talked about aerosol climate effects, I was asking if you have taken this into account. If not, you should make caveats and show a whole picture to avoid misleading messages.

**Response:** We sincerely thank the reviewer for the positive comment on our revised manuscript. We did not take the lifetime of cloud into account during the calculation. We added the caveats to specify the conditions, under which our results were derived, in the discussion part in Page 13 Line 17-18.

"Note that the increase of $N_c$ could enhance the evaporation and further reduce the lifetime of cloud, which was not taken into account when calculating the induced albedo."

Responses to the editor's comments:

Comments to the Author:
The authors have addressed most of the reviewer comments. There are still a number of points that need to be corrected for before publication in ACP. Please address the important point raised by Reviewer #1 (see the attached report) and the following issues that I discovered in an editorial review of the manuscript:

We sincerely thank the editor for the valuable comments and suggestions. We have responded the editor's comments one by one in the following. Comments by the editor are given in black normal font, and our response to the comments is shown in blue. Newly added and modified text in the revised manuscript is given in red.

**Comment 1**
- The manuscript still needs to be proofread and corrected for grammatical errors. Please make sure this is done before the final submission.
**Response:** Thank you for the comment. We have proofread our manuscript and corrected the grammatical errors.

**Comment 2**
- Page 6, line 8: Please add "here" between "which" and "represent"
**Response:** We have revised this sentence as you suggested. (Page 6 Line 6-7)
"Aerosol indirect effect (AIE), which here represents simply approximations of the derivatives of the cloud microphysics ($r_{eff}$ and $N_C$) with respect to changes in aerosol concentrations"

**Comment 3**
- Please make sure you cite all figures in the order in which they are presented. Revise the order of figures, if need be.
**Response:** We have rearranged and rechecked the order of the tables and figures. They are now cited in the order.

**Comment 4**
- P. 7, lines 10-21: Please do not refer to the cloud events before they are introduced.
**Response:** We rechecked about this through the whole manuscript. We make sure that cloud events were first introduced (expressed as CE-xx) and then be referred to.

**Comment 5**
- P. 7, lines 27-28: Please state more explicitly (with the relevant numerical values) why this was the case and how the calculations were made with appropriate references to the supplement.
**Response:** We explained the calculation with the numerical values and revised more detailed in Page 7 Line 24-28.
"The detailed information could be found in the Supplement (Table S2, Fig. S3 and Fig. S4) and was briefly introduced here. The sampling angle ($\theta_s$) and $v_{up}$ for CP-1 and CP-2 were 11.9 ° and 0.82 ± 0.29

m s$^{-1}$, and 10.6 ° and 0.92 ± 0.36 m s$^{-1}$, respectively (Table S2). According to the calculations provided by (Spiegel et al., 2012), the aspiration efficiency and transmission efficiency were all close to 1."

**Comment 6**

- P. 8, lines 26-28: Since causality cannot be proven based on just the data, please replace "would cause" with "is observed to coincide".

**Response:** Thank you for the comment. This sentence was revised as "In S4, the increase of PM$_{2.5}$, through evaporation of cloud droplets or lifting of *CBH* (Fig. 2), was observed to coincide with the vanishment of cloud events (Mazoyer et al., 2019;Li et al., 2017a)." (Page 8 Line 24-26)

**Comment 7**

- Section 3.2.1: Please carefully go through and revise this section to accurately describe and distinguish covariation and causality. Most of the data presented show covariations (or not) between different variables, but causality is difficult to prove.

**Response:** We have revised the Section 3.2.1. Consistent variation and inverse variation between $N_P$ and $N_C$ were clearly pointed and described. We weakened some statements and expressed the possible reasons according to our results.

**Comment 8**

- P. 11, line 3: Isn't this paragraph somewhat contradictory to what has been said before? Please revise.

**Response:** We made a mistake in the equation of $AIE_N$ in section 2.7 in the last version, that the negative sign should be removed. This mistake may cause the contradictory. We have corrected the equation in the revised manuscript. The description is not contradict, but we have changed the order of the description in section 3.2.2 to make this part more clear. The positive values of *AIEr* and $AIE_N$ represent that $N_C$ increases with the decrease of $r_{eff}$ and the increase of $N_P$, and vice versa. During CP-1 and CP-2, no negative values of *AIEr* and $AIE_N$ were observed, while the specific values of *AIEr* and $AIE_N$ during CP-1 and CP-2 were different.

**Comment 9**

- P. 14, Section 4: Please revise also this section to accurately describe covariation vs. causality in the data set.

**Response:** We have revised the conclusion part and clearly pointed and described about the covariation and the causality. We weakened some statements and expressed the possible reasons according to our results.

---

## Author Response (AR4)

Dear Editor,

We sincerely thank the comments from you and sorry for language mistakes in the manuscript. In this version, we have carefully corrected the issues related to the language. Except making the suggested changes, other issues we found were also marked in red in the revised manuscript and Supplement Information. Please check. Thank you so much.

Best regards,

Jianmin Chen